# The Sectional Stratospheric Sulfate Aerosol module S3A-v1 within the LMDZ general circulation model: Description and evaluation against stratospheric aerosol observations

Christoph Kleinschmitt[1], Olivier Boucher[2], Slimane Bekki[3], François Lott[4], and Ulrich Platt[1]

[1]Institute of Environmental Physics, Heidelberg University, Im Neuenheimer Feld 229, 69120 Heidelberg, Germany
[2]Institut Pierre-Simon Laplace, CNRS / UPMC, 4 place Jussieu, 75252 Paris Cedex 05, France
[3]Laboratoire Atmosphères Milieux Observations Spatiales, Institut Pierre-Simon Laplace, CNRS / UVSQ, 11 boulevard d'Alembert, 78280 Guyancourt, France
[4]Laboratoire de Météorologie Dynamique, Institut Pierre-Simon Laplace, CNRS / ENS, 24 rue Lhomond, 75231 Paris Cedex 05, France

*Correspondence to:* Christoph Kleinschmitt (christoph.kleinschmitt@iup.uni-heidelberg.de)

**Abstract.** Stratospheric aerosols play an important role in the climate system by affecting the Earth's radiative budget as well as atmospheric chemistry and the capabilities to simulate them interactively within global models are continuously improving. It is important to represent accurately both aerosol microphysical and atmospheric dynamical processes because together they affect the size distribution and the residence time of the aerosol particles in the stratosphere. The newly developed LMDZ-S3A model presented in this article uses a sectional approach for sulfate particles in the stratosphere and includes the relevant microphysical processes. It allows full interaction between aerosol radiative effects (e.g. radiative heating) and atmospheric dynamics including e.g. an internally-generated quasi-biennial oscillation (QBO) in the stratosphere. Sulfur chemistry is semi-prescribed via climatological lifetimes. LMDZ-S3A reasonably reproduces aerosol observations in periods of low (background) and high (volcanic) stratospheric sulfate loading, but tends to overestimate the number of small particles and to underestimate the number of large particles. Thus, it may serve as a tool to study the climate impacts of volcanic eruptions, as well as the deliberate anthropogenic injection of aerosols into the stratosphere, which has been proposed as a method of geoengineering to abate global warming.

## 1 Introduction

The study of stratospheric aerosols has traditionally been a separate activity to that of tropospheric aerosols, *inter alia* because of different observing methods and observing systems. This has also been true for the modelling efforts because, due to different residence times of aerosols in the troposphere and stratosphere, the relevance and relative importance of the various processes at play are different. Resolving accurately the size distribution of aerosol particles is crucial to calculate correctly the lifetime, vertical distribution and radiative properties of aerosol particles in the stratosphere, whereas tropospheric aerosol models can in first approximation rely on the assumption of self-preserving modes in the aerosol size distribution. Gravitational sedimentation, which is the main loss process for aerosols in the stratosphere (Deshler, 2008), is extremely dependent on the

size of the aerosol particles. Coagulation, a fairly non-linear process, is also dependent on the details of the aerosol size distribution. The importance of resolving accurately the size distribution, notably the large particles tail of the distribution where most the sedimentation mass flux takes place, was already identified in the early modelling studies (Turco et al., 1979; Pinto et al., 1989), which used sectional aerosol models with a relatively high resolution in aerosol size but only one dimension (i.e. height) in space.

An accurate representation of dynamical processes in the stratosphere (e.g., subtropical meridional transport barriers, Brewer-Dobson circulation, stratosphere-troposphere exchange) is also paramount to properly simulate the distribution of stratospheric aerosols and their dispersion following volcanic eruptions. Representing accurately the interplay between aerosol microphysical and dynamical processes can be presumed to be computationally very expensive as it involves at least five dimensions: three space dimensions, the aerosol size dimension, and the time dimension. This means that, for a given computational cost, some trade off is necessary between the representation (or discretization) of these dimensions and/or the length of the simulation and the number of simulations. One possibility is to exploit the approximately zonal symmetry in the stratosphere to reduce the atmosphere to the height and latitude dimensions. This approach was used in particular by Bekki and Pyle (1992) and Mills et al. (1999) who retained the sectional approach to represent aerosol size.

The increase in computational capability has progressively allowed the development of three-dimensional models of stratospheric aerosols in the late 1990s / early 2000s. Most of these models were initially chemistry-transport models (CTMs). This so-called offline approach was often preferred because chemistry-transport models are cheaper to run as wind and temperature fields are specified according to meteorological analyses instead of being calculated prognostically like in climate-chemistry models. In addition, since the transport of tracers is driven by meteorological analyses, the observed day-to-day variability in chemical composition can be reproduced (at least to some extent), facilitating the comparisons with measurements. While chemistry-transport models are suitable for a broad range of studies, they do not include any radiative feedback between chemical composition and dynamics, and notably ignore the radiative effects of aerosols onto atmospheric dynamics.

Given the importance of stratospheric aerosols for the Earth's radiative budget, there is also a need to represent stratospheric aerosols in climate models. The volcanic aerosol forcing is important to simulate the temporal evolution of the climate system over the last millennium in general, and over the instrumental period (1850 to present-day) in particular. This was initially done by prescribing the amount and properties of stratospheric aerosols as (time-varying) climatologies derived from observations. This was the case in most if not all of the climate models involved in the 5th phase of the Climate Model Intercomparison Project (CMIP5) as discussed in Flato et al. (2013), and it is still expected to be the case in the forthcoming sixth phase (CMIP6) (Eyring et al., 2016). However capabilities to simulate stratospheric aerosols within global climate models are continuously improving.

The SPARC stratospheric aerosol assessment report (Thomason and Peter, 2006) provides a review of stratospheric aerosol models as of 10 years ago. It clearly represents a milestone and stimulated significant further model development since then. As a result, the recent review by Kremser et al. (2016) lists more than a dozen global three-dimensional stratospheric aerosol models. It should be noted however that several of these configurations share the same atmospheric general circulation model (GCM) or the same aerosol module and not all of them include the interaction of aerosols with radiation.

The sectional approach has been adopted by a number of these three-dimensional stratospheric aerosols (e.g., Timmreck, 2001; Pitari et al., 2002; Sheng et al., 2015). Stratospheric aerosols have also been modelled in climate models as an extension of schemes initially designed for tropospheric aerosols. Simple mass-based (i.e., bulk) aerosol schemes modified to account for gravitational settling of the sulphate aerosols have been used occasionally (Oman et al., 2006; Haywood et al., 2010; Aquila et al., 2012). Such models do not represent the growth of aerosol particles but rely instead on a fixed size distribution for each aerosol type. More sophisticated approaches have also been developed, whereby the aerosol size distribution is approximated by a statistical function with a pre-defined shape and a few variable parameters. The evolution of the size distribution is governed through variations in these selected parameters but, by construction, it has only a few degrees of freedom, which may lead to discrepancies and artefacts in the simulated size distribution. Examples include two-moment modal aerosol microphysics schemes such as the M7 model (Vignati et al., 2004; Stier et al., 2005) and the GLOMAP model (Mann et al., 2010; Dhomse et al., 2014), whereby each aerosol mode is represented prognostically by a number and a mass concentration.

A key question relates to the performance of the different approaches for representing the aerosol size distribution. Weisenstein et al. (2007) compared sectional and modal aerosol schemes. They found that the modal aerosol schemes performed adequately against the sectional aerosol schemes for aerosol extinction and surface area density but less so for effective radius. Kokkola et al. (2009) found considerable deviations in the simulated aerosol properties between sectional and modal aerosol schemes for elevated $SO_2$ concentrations but they focused on very short timescales after a $SO_2$ burst, and it could be that the discrepancy is less on longer timescales. The modal schemes have the advantage of being computationally cheap (relative to the sectional schemes), but may have to be "tuned" against results of the sectional scheme. The sectional approach has the advantage that the number of size bins can be increased to increase the accuracy of the aerosol scheme. If the scheme is numerically stable, it should converge to a (numerical) solution when the number of size bins increases. It is thus possible to evaluate the uncertainty induced by limiting the number of size bins, whereas it is difficult, if not impossible, in the modal approach to assess the uncertainty induced by the assumption of pre-defined aerosol modes with pre-defined shape. This does not mean however that aerosol sectional scheme will always be superior, as in the end, it will be subject to the same computational trade off as other models, and the relatively large cost of the sectional approach may limit the horizontal or vertical resolutions of the atmospheric model.

A climate model with a well established stratospheric aerosol capability is the WACCM/CARMA model described by English et al. (2011) (for WACCM see Garcia et al. (2007) and for CARMA see Toon et al. (1988)) which includes a sectional stratospheric aerosol with all the relevant chemistry and microphysics, along with a high vertical resolution. However the model does not consider aerosol radiative heating. Recently Mills et al. (2016) used the WACCM model to simulate the time evolution of the stratospheric aerosol over the period 1990–2014. They find a good agreement in stratospheric aerosol optical depth (SAOD) with SAOD derived from several available lidar measurements by Ridley et al. (2014) and in surface area density (SAD) with balloon-borne optical particle counter (OPC) measurements at the University of Wyoming (Kovilakam and Deshler, 2015).

Our research on stratospheric aerosols is motivated by their interaction with both incoming solar radiation and outgoing terrestrial radiation, and the associated climate response to such a radiative forcing. We are interested in a wide range of

stratospheric aerosol burdens: from background levels to the large volcanic loads observed after major eruptions such as El Chichón, Pinatubo (e.g., Dutton and Christy, 1992), Krakatoa, or Tambora. Furthermore we are interested in studying the potential of stratospheric aerosol injection (SAI) as a geoengineering mean to artificially cool the Earth's climate in order to compensate for greenhouse gas global warming (e.g., Budyko, 1977; Crutzen, 2006). Recent reviews of scientific studies and open questions regarding solar geoengineering (e.g., Irvine et al., 2016; MacMartin et al., 2016) highlighted again the need for accurate stratospheric aerosol models. These research interests motivate the introduction of a versatile stratospheric aerosol model within the IPSL Climate Model (IPSL-CM) and its atmospheric component LMDZ. As a first step towards this objective we have introduced a sectional stratospheric sulfate aerosol model in the LMDZ model. We have included processes relevant to both the background and volcanic stratospheric aerosol layer, but also processes relevant to much larger and/or longer emission rates than experienced in typical volcanic eruptions.

In this article we offer a full and detailed description of the aerosol model in Sect. 2. We also evaluate its performance against available observations. The eruption of Mount Pinatubo in June 1991 is the last major eruption experienced by the Earth and was relatively well observed. As such it is a useful case study for any stratospheric aerosol model and is discussed in Sect. 3.

## 2 Model description

The newly developed sectional stratospheric sulfate aerosol (S3A) module is now part of the three-dimensional atmospheric general circulation model (GCM) LMDZ described in Hourdin et al. (2006) and Hourdin et al. (2013). LMDZ itself can be coupled to the ORCHIDEE land surface model (Krinner et al., 2005), the oceanic GCM NEMO (Madec, 2008) and other biogeochemical or chemical model components to form the IPSL Earth System Model (Dufresne et al., 2013). It is thus possible to use the S3A model to study the climate response to volcanic eruptions or SAI. We briefly describe below the host atmospheric model in Sect. 2.1 and make a comprehensive description of the S3A model in Sect. 2.2.

### 2.1 Host atmospheric model

#### 2.1.1 Model resolution and model physics

A full description of the LMDZ model in its LMDZ5A configuration is available in Hourdin et al. (2006) and Hourdin et al. (2013). We do not repeat the description here but instead focus on the evolutions of the model since Hourdin et al. (2013) and the specificities of the LMDZ configuration considered in this study.

In the configuration tested here with the S3A module, LMDZ is run with 96×96 grid-points, i.e. a horizontal resolution of 1.89° in latitude and 3.75° in longitude –which is the same as for LMDZ5A–, but with a vertical resolution increased to 79 layers and a model top height of 75 km. The additional layers are mostly located in the stratosphere so that in the lower stratosphere (between 100 and 10 hPa) the vertical spacing $\Delta z$ is approximately 1 km in this model setup. The increased resolution on the vertical aims to "close" the stratospheric circulation. It is also necessary to generate a realistic quasi-biennial

oscillation (QBO) as discussed below. de la Cámara et al. (2016) provide a more extensive description of the stratospheric dynamics modelled with this vertically enhanced configuration of LMDZ.

Our configuration of the LMDZ model differs from that described in Hourdin et al. (2013) in that it has a different radiative transfer code. In the shortwave, the code is an extension to 6 bands of the initial 2-band code that is used in LMDZ5A (Fouquart and Bonnel, 1980), as implemented in a previous version of the ECMWF numerical weather prediction model. In the longwave, we use the ECMWF implementation of the RRTM radiative transfer scheme (Mlawer et al., 1997) with 16 spectral bands. This change in radiative transfer scheme is motivated by the necessity to account for the radiative effects of the stratospheric aerosols both in the shortwave (SW) and longwave (LW) part of the spectrum with sufficient spectral resolution.

Finally it should be noted that the time step for the model physics, $\Delta t_{\mathrm{phys}}$, is unchanged at 30 minutes, which is also the main timestep used for the S3A model.

### 2.1.2  Tropopause recognition

As the model focuses on stratospheric aerosols, the separation between troposphere and stratosphere (i.e. the location of the tropopause) is of special importance. The S3A model requires the knowledge of whether a particular model grid box is located in the troposphere or the stratosphere, because the processes of nucleation, condensation, evaporation and coagulation are only activated in the stratosphere. Tropospheric aerosols are treated separately by a standard bulk aerosol model (e.g., Escribano et al., 2016). Also we have a set of stratospheric aerosol variables that are only diagnosed in the stratosphere.

To this effect we use the algorithm by Reichler et al. (2003) which is based on the WMO definition of the tropopause as "the lowest level at which the lapse-rate decreases to $2\,\mathrm{K\,km^{-1}}$ or less, provided that the average lapse-rate between this level and all higher levels within 2 km does not exceed $2\,\mathrm{K\,km^{-1}}$". We use the FORTRAN code provided by Reichler et al. (2003) which we have adapted to the LMDZ model. With this the tropopause pressure is computed at each timestep. In the rare case that the algorithm does not find the tropopause in a grid column, it is set to a default value that only depends on the latitude $\varphi$ (in radians) through the relationship:

$$p_{\mathrm{TP}}\,[\mathrm{hPa}] = 500 - 200 \cdot \cos(\varphi) \tag{1}$$

In this case the tropopause is assumed to vary between $300\,\mathrm{hPa}$ at the Equator to $500\,\mathrm{hPa}$ at the Poles, independently of the season.

### 2.1.3  Quasi-biennial oscillation in the stratosphere

The vertical extension to the LMDZ domain, as discussed above, is accompanied by a new stochastic parametrisation of gravity waves produced by convection which is documented in Lott and Guez (2013). This is another difference to the original LMDZ5A model configuration described in Hourdin et al. (2013) which did not include this parametrisation. The combination of the extended vertical resolution in the lower stratosphere and the gravity wave parametrisation generates a QBO in the model, as shown in Fig. 1 and documented in Lott and Guez (2013). The amplitude of the QBO around 10 hPa is around $10\text{--}15\,\mathrm{m\,s^{-1}}$ and is smaller than observed ($20\text{--}25\,\mathrm{m\,s^{-1}}$). The easterly phases are also stronger and longer in duration than the westerly

phases, which is realistic. One subtle difference with the QBO shown in Lott and Guez (2013) is that here the connection with the semi-annual oscillation (SAO) above is quite pronounced, whereas it was not so evident in Lott and Guez (2013). This is because the characteristic phase speed we have recently adopted for the convective gravity waves $C_{\max} = 30\,\mathrm{m\,s^{-1}}$ (de la Cámara et al., 2016) is smaller than for the convective gravity waves parameter we used in Lott and Guez (2013). The stronger connection can be further explained by the fact that more waves travelling through the QBO sector will likely be absorbed by the critical levels produced by the SAO. Finally, it is worthwhile to recall that our QBO does not extend down to 100 hPa, in contradiction with observations. In our model, it is probably due to the fact that we underestimate the explicit slow Kelvin waves that play a crucial role in the lower stratosphere (Giorgetta et al. (2006), for the Kelvin waves in models, see Lott et al. (2014)).

It is noticeable that the period of our simulated QBO in Fig. 1 shortens and its amplitude increases during the second half of the simulation, e.g. as the aerosol layer builds up in the lower stratosphere. More precisely, the QBO has a period well above 26 months during the first five years before evolving to an almost purely biennial oscillation by the end of the simulation. This could be due to the warming of the stratosphere induced by the developing stratospheric aerosol layer (up to 1.5 K in the tropical lower stratosphere), which would be consistent with the opposite behaviour found when the stratosphere cools, e.g. in response to an increase in greenhouse gases (for the intensity and period see de la Cámara et al. (2016), for intensity only, see the observations in Kawatani and Hamilton (2013)). As our simulation is quite short, this result should be consolidated by longer runs. It should also be kept in mind that since the QBO in our model is probably oversensitive to changes in greenhouse gases, it may also be oversensitive to the aerosol content.

Despite these shortcomings, the self-generated QBO is an attractive feature of the LMDZ model to study stratospheric aerosols and different SAI scenarios. A more realistic simulation of the QBO would require a higher horizontal resolution.

### 2.1.4 Nudging to meteorological reanalysis

As an option, the LMDZ model can be nudged to a meteorological reanalysis. This is useful to simulate a historical situation with particular meteorological conditions. Only the horizontal wind components $u$ and $v$ are nudged. Nudging is performed by adding an additional term to the governing differential equations for $u$ and $v$ which relaxes the wind towards a meteorological reanalysis:

$$\frac{\partial u}{\partial t} = \frac{\partial u}{\partial t}_{\mathrm{GCM}} + \frac{u_{\mathrm{reanalysis}} - u}{\tau} \qquad ; \qquad \frac{\partial v}{\partial t} = \frac{\partial v}{\partial t}_{\mathrm{GCM}} + \frac{v_{\mathrm{reanalysis}} - v}{\tau} \tag{2}$$

where the relaxation time $\tau$ is taken to 30 minutes. Nudging is activated in the model calculations described in Sect. 3.2 using the ERA-Interim reanalysis reprojected onto the LMDZ grid. In this case, the reanalysed QBO prevails over the model self-generating QBO described in the previous section.

## 2.2 The sectional stratospheric sulfate aerosol (S3A) model

### 2.2.1 Prognostic variables

The S3A module in the configuration introduced here represents the stratospheric aerosol size distribution with $N_B = 36$ size bins of sulfate particles with dry radius ranging from 1 nm to 3.3 µm (i.e., $r_1 = 1\,\text{nm}$ and $r_N = 3.3\,\text{µm}$ for particles at 293 K consisting of 100 % $H_2SO_4$) and particle volume doubling between successive bins (i.e., $R_V = V_{k+1}/V_k = r_{k+1}^3/r_k^3 = 2$ for $1 \le k < N_B$). The number of size bins $N_B$ and the corresponding value of $R_V$ represent a compromise between the accuracy of the scheme, which increases with higher resolution in size, and the computational cost of the model.

It should be noted that the $r_k$ are the radius of the "middle" of the size bins. The radii of the lower and upper boundaries of bin $k$ are

$$r_k^{\text{lower}} = \begin{cases} r_k/\sqrt{R_V^{1/3}} & \text{for } k = 1 \\ \sqrt{r_{k-1}\,r_k} & \text{for } 1 < k \le N_B \end{cases} \quad ; \quad r_k^{\text{upper}} = \begin{cases} \sqrt{r_k\,r_{k+1}} & \text{for } 1 \le k < N_B \\ r_k \cdot \sqrt{R_V^{1/3}} & \text{for } k = N_B \end{cases} . \tag{3}$$

Other global stratospheric models with sectional aerosol schemes have resolutions ranging from 11 to 45 size bins (Thomason and Peter, 2006). Our model resolution of 36 size bins is therefore on the high end of this range. A relatively high size resolution is required for an accurate modelling of stratospheric aerosols because coagulation, an important process in the stratosphere, and gravitational sedimentation, the main loss process in the stratosphere, are very strongly dependent on the aerosol size.

The lower end of our size range (1 nm) was chosen to be close to the size of typical freshly-nucleated particles. We have tried to limit the number of bins by increasing the minimum aerosol size to 10 nm or more and feeding the nucleation term directly into this bin. However this resulted in inaccuracies in the size distribution both at small and large aerosol sizes, so this simplification was eventually not adopted. Large particles have short residence times and therefore very low concentrations in the stratosphere. As a result they do not contribute much to the aerosol optical depth, hence it is acceptable to set an upper range to 3.3 µm for our modelled size distribution. While 36 size bins correspond to our current configuration, the size range and size resolution can easily be changed in our model by adjusting the number of size bins ($N_B$), the minimum dry aerosol size ($r_1$), and the volume ratio between bins ($R_V$). All the parametrisations described below then adjust to the new size discretisation.

Aerosol amount in each of the size bins is treated as a separate tracer for atmospheric transport in the unit of particle number per unit mass of air as required by our mass-flux scheme (Hourdin and Armengaud, 1999).

### 2.2.2 Semi-prognostic sulfur chemistry

Besides the concentrations of aerosol particles in each bin, the module also represents the sulfate aerosol precursor gases OCS and $SO_2$ as semi-prognostic variables and gaseous $H_2SO_4$ as a fully-prognostic variable. The mass mixing ratios of OCS and $SO_2$ are initialised to climatological values at the beginning of a simulation. They are also prescribed throughout the simulation to climatological values below 500 hPa, but they evolve freely above that pressure level where they are subject to advection, convective transport, wet deposition, and chemical transformations. The chemical reactions transforming one species to another

(OCS into SO$_2$ and SO$_2$ into H$_2$SO$_4$) during one model timestep are parametrised as exponential decay terms with prescribed chemical lifetimes:

$$\Delta[\text{SO}_2] \quad = \frac{M(\text{SO}_2)}{M(\text{OCS})} \cdot [\text{OCS}] \cdot \left[1 - \exp\left(-\frac{\Delta t_{\text{phys}}}{\tau_{\text{OCS}}}\right)\right] \tag{4}$$

$$\Delta[\text{H}_2\text{SO}_4] = \frac{M(\text{H}_2\text{SO}_4)}{M(\text{SO}_2)} \cdot [\text{SO}_2] \cdot \left[1 - \exp\left(-\frac{\Delta t_{\text{phys}}}{\tau_{\text{SO}_2}}\right)\right] \tag{5}$$

5 with $[X]$ being the mixing ratio, $\Delta[X]$ the change in mixing ratio, $M(X)$ the molecular mass and $\tau_X$ the chemical lifetime of species $X$.

Both the climatological values of OCS and SO$_2$ and their chemical lifetimes are taken from a latitude-altitude climatology at monthly resolution from the UPMC/Cambridge global two-dimensional chemistry-aerosol-transport model (Bekki and Pyle, 1992, 1993). These quantities are shown in Fig. 2. Using prescribed chemical lifetimes means that the OCS and SO$_2$ 10 concentrations do not feed back onto concentrations of oxidants which oxidise these species. This is a limitation of our model, especially in situation of large OCS or SO$_2$ injection rates, which we discuss further in Sect. 3.3.2. In a future study the S3A model will be coupled to the REPROBUS (Reactive Processes Ruling the Ozone Budget in the Stratosphere) model for stratospheric chemistry that is also available in the LMDZ model (Lefèvre et al., 1994, 1998).

A schematic of the model species and physical processes is shown in Fig. 3. The following processes are represented: aerosol 15 nucleation from gaseous H$_2$SO$_4$, condensation and evaporation of gaseous H$_2$SO$_4$, coagulation and sedimentation of aerosol particles. Dry and wet deposition of gas-phase species and aerosols in the troposphere are also considered as we are interested in the tropospheric fate of the stratospheric aerosols.

### 2.2.3 Nucleation

The formation rate of new particles via binary homogeneous nucleation of sulfuric acid and water vapour is parametrised 20 as a function of the sulfuric acid gas phase concentration, the relative humidity and the absolute temperature as described by Vehkamäki et al. (2002). This parametrisation provides the nucleation rate $J_{\text{nuc}}$ in unit of particles cm$^{-3}$ s$^{-1}$, the total number of molecules in each nucleated particle $N_{\text{tot}}$ and the mole fraction of H$_2$SO$_4$ in the new particle $x$. The equations are cumbersome and not repeated here, but it should be noted that we rely on the Fortran code provided by Vehkamäki et al. (2002).

25 The parametrisation is not valid any more under conditions where the number of molecules in the critical cluster is below 4 (which occurs mainly at large H$_2$SO$_4$ vapour concentrations). Under such conditions, we take the collision rate of two H$_2$SO$_4$ molecules as the nucleation rate instead [Hanna Vehkamäki, personal communication], i.e. $N_{\text{tot}} = 2$, $x = 1$ and

$$J_{\text{nuc}}(N_{\text{tot}} < 4) = [\text{H}_2\text{SO}_4]^2 \cdot \left(\frac{3\pi}{4}\right)^{1/6} \cdot \left(\frac{12 k_{\text{B}} T}{M(\text{H}_2\text{SO}_4)}\right)^{1/2} \cdot (2 V(\text{H}_2\text{SO}_4)^{1/3})^2 \tag{6}$$

with $M(\text{H}_2\text{SO}_4)$ the molecular mass of sulfuric acid, $[\text{H}_2\text{SO}_4]$ the concentration of H$_2$SO$_4$ (in molecules cm$^{-3}$), $k_{\text{B}}$ the Boltz-30 mann constant, and $V(\text{H}_2\text{SO}_4)$ the molecular volume of H$_2$SO$_4$ which is computed using Vehkamäki's density parametrisation.

In order to sort the new particles into the model size bins in a mass-conserving way, their volume is computed as

$$V_{\text{new}}^{\text{nuc}} = \frac{M(\text{H}_2\text{SO}_4) \, N_{\text{tot}} \, x}{\rho(\text{H}_2\text{SO}_4)} \tag{7}$$

with the density of sulfuric acid $\rho(\text{H}_2\text{SO}_4)$ taken at the reference temperature 293 K. The new particles are distributed among the size bins using a method inspired by the distribution factor $f_{i,j,k}$ from Jacobson et al. (1994) described in Sect. 2.2.6 and Eq. (19). Hereby for each new particle we add $f_k^{\text{nuc}} \cdot V_{\text{new}}^{\text{nuc}}/V_k$ particles to bin $k$, with $V_k = \frac{4}{3}\pi r_k^3$ and

$$f_k^{\text{nuc}} = \begin{cases} \left(\dfrac{V_{k+1} - V_{\text{new}}^{\text{nuc}}}{V_{k+1} - V_k}\right) \dfrac{V_k}{V_{\text{new}}^{\text{nuc}}} & \text{for } V_k \leq V_{\text{new}}^{\text{nuc}} < V_{k+1} \; ; \; k < N_B \\ 1 - f_{k-1}^{\text{nuc}} & \text{for } V_{k-1} \leq V_{\text{new}}^{\text{nuc}} < V_k \; ; \; k > 1 \\ 1 & \text{for } [V_{\text{new}}^{\text{nuc}} \leq V_k \; ; \; k = 1] \text{ or } [V_{\text{new}}^{\text{nuc}} \geq V_k \; ; \; k = N_B] \\ 0 & \text{otherwise} \end{cases} \tag{8}$$

As a result the actual particle nucleation rate may deviate from Vehkamäki's parametrised value. For example, if the nucleated particles have only half the volume of a particle in the smallest model size bin, the number of new particles is only half of the parametrised value, but the $\text{H}_2\text{SO}_4$ flux from the gas to the particle phase is the same. We favoured conserving sulfur mass over conserving particle number concentration. This approximation on the exact value of the nucleation rate is not expected to have a very significant impact on the results because the particle size distribution is mainly determined by coagulation and condensation (English et al., 2011). Furthermore this approximation is justified in the light of the large uncertainties arising from parametrizing nucleation rates using grid-box quantities (temperature, $\text{H}_2\text{O}$, $\text{H}_2\text{SO}_4$) that neglect sub-grid scale variations.

### 2.2.4 Condensation and evaporation of sulfuric acid

The change in size of the sulfate particles through gain from or loss towards the $\text{H}_2\text{SO}_4$ gas phase is computed based on the UPMC / Cambridge model parametrisation (Bekki and Pyle, 1992, 1993).

First the saturation vapor pressure of $\text{H}_2\text{SO}_4$ over a flat surface is calculated from a relationship given by Ayers et al. (1980) using the values of $\text{H}_2\text{SO}_4$ chemical potentials in aqueous phase listed in the work of Giauque et al. (1960, Table I):

$$p_{\text{H}_2\text{SO}_4}^{\text{sat}} \, [\text{Pa}] = \frac{101325}{0.086} \cdot \exp\left(-\frac{10156}{T} + 16.259 + \frac{\mu - \mu_0}{\text{R} \, T}\right) \tag{9}$$

with $T$ the temperature, $\mu$ the chemical potential and R the ideal gas constant. However, as recommended in Hamill et al. (1982), vapour pressures of $\text{H}_2\text{SO}_4$ from the Ayers et al. (1980) relationship are divided by 0.086 to obtain values close to the measurements of Gmitro and Vermeulen (1964).

We account for the Kelvin effect, whereby the saturation vapour pressure of $\text{H}_2\text{SO}_4$ over a curved surface is higher than the saturation vapour pressure over a flat surface. The saturation vapour pressure over a sulfate aerosol particle in size bin $k$, with radius $r_k$, is:

$$p_{\text{H}_2\text{SO}_4}^{\text{sat},k} = p_{\text{H}_2\text{SO}_4}^{\text{sat}} \cdot \exp\left(\frac{2\,\sigma\,M(\text{H}_2\text{O})}{\rho_\text{p}\,k_\text{B}\,T\,r_k}\right) \tag{10}$$

with $M(H_2O)$ the molecular mass of water, $\sigma$ the surface tension of the sulfuric acid solution (which is set to $72\,\mathrm{mN\,m^{-1}}$, the value for water at $20°\,\mathrm{C}$) and $\rho_p$ the density of the sulfate particles. The corresponding $H_2SO_4$ number density at saturation is then

$$[H_2SO_4]^{\mathrm{sat},k} = \frac{p_{H_2SO_4}^{\mathrm{sat},k}}{k_B T} \tag{11}$$

5    Then the flux of $H_2SO_4$ between the particle and the gas phase, $J_k(H_2SO_4)$, in molecules $\mathrm{particle^{-1}\,s^{-1}}$ is computed individually for every size bin $k$ following Seinfeld and Pandis (2006, p. 542-547)

$$J_k(H_2SO_4) = \pi\,r_k^2\,\bar{v}(H_2SO_4)\,\alpha\cdot\left([H_2SO_4] - [H_2SO_4]^{\mathrm{sat},k}\right)\cdot\frac{1+\mathrm{Kn}_k}{1+\mathrm{Kn}_k+\alpha/(2\,\mathrm{Kn}_k)} \tag{12}$$

with the molecular accommodation coefficient $\alpha = 0.1$, $\bar{v}(H_2SO_4)$ the thermal velocity of a $H_2SO_4$ molecule and $\mathrm{Kn}_k = \lambda/r_k$ the Knudsen number, where we use the parametrisation from Pruppacher and Klett (2010, p. 417) for the mean free path of air

$$10 \quad \lambda = \lambda_0\cdot\left(\frac{p_0}{p}\right)\cdot\left(\frac{T}{T_0}\right) \tag{13}$$

with $\lambda_0 = 6.6\cdot 10^{-8}\,\mathrm{m}$ for air at standard conditions $p_0 = 1013.25\,\mathrm{hPa}$ and $T_0 = 293.15\,\mathrm{K}$.

Evaporation from a particle over one time step is limited to its actual $H_2SO_4$ content and condensation is limited by the available $H_2SO_4$ vapour. How this is dealt with is further described in Sect. 2.2.5.

Condensation (evaporation) has an impact on the particle size distribution, shifting particles to larger (smaller) size. To

15    account for this, we first compute the new particle volume after adding the flux $J_k(H_2SO_4)$ over the timestep $\Delta t$:

$$V_{k,\mathrm{new}}^{\mathrm{c/e}} = V_k\cdot\left(1 + \frac{J_k(H_2SO_4)\Delta t}{N_k(H_2SO_4)}\right) \tag{14}$$

where $N_k(H_2SO_4)$ is the number of sulfuric acid molecules in a particle for bin $k$. Knowing this new volume of a particle coming from bin $k$ and experiencing condensation or evaporation, the distribution among all the size bins (index $l$) can then be computed analogously to Eq. (8) and (19) using a factor:

$$20 \quad f_{k,l}^{\mathrm{c/e}} = \begin{cases} \left(\dfrac{V_{l+1} - V_{k,\mathrm{new}}^{\mathrm{c/e}}}{V_{l+1} - V_l}\right)\dfrac{V_l}{V_{k,\mathrm{new}}^{\mathrm{c/e}}} & \text{for } V_l \leq V_{k,\mathrm{new}}^{\mathrm{c/e}} < V_{l+1}\,;\, l < N_B \\ 1 - f_{k,l-1}^{\mathrm{c/e}} & \text{for } V_{l-1} \leq V_{k,\mathrm{new}}^{\mathrm{c/e}} < V_l\,;\, l > 1 \\ 1 & \text{for } \left[V_{k,\mathrm{new}}^{\mathrm{c/e}} \leq V_l\,;\, l = 1\right] \text{ or } \left[V_{k,\mathrm{new}}^{\mathrm{c/e}} \geq V_l\,;\, l = N_B\right] \\ 0 & \text{otherwise} \end{cases} \tag{15}$$

### 2.2.5   Competition between nucleation and condensation

As both processes, nucleation and condensation, consume $H_2SO_4$ vapour while having very different effects on the particle size distribution, the competition between the two processes has to be handled carefully in a numerical model. Furthermore this has to be done at an affordable numerical cost as we aim to perform long global simulations. We address this in the

S3A module using an adaptive sub-timestepping. After computing the $H_2SO_4$ fluxes due to nucleation and condensation in $kg\,H_2SO_4\,s^{-1}$ from the initial $H_2SO_4$ mixing ratio, a sub-timestep, $\Delta t_1$, is computed such that the sum of both the nucleation and condensation fluxes consumes no more than 25% of the available ambient $H_2SO_4$ vapour:

$$\Delta t_1 = \min \left( 0.25 \cdot \frac{[H_2SO_4]_0}{J_{\text{nuc}} + J_{\text{cond}}} \,,\, \Delta t_{\text{phys}} \right) \tag{16}$$

where $\Delta t_{\text{phys}}$ is the main timestep (30 minutes in our case), and $[H_2SO_4]_0$ is the $H_2SO_4$ mixing ratio at the beginning of the timestep. Hence, neither one of the two processes can use up all the sulfuric acid at the expense of the other process. This sub-timestepping procedure is repeated up to 4 times with a sub-timestep equal to

$$\Delta t_{1<i<4} = \min \left( 0.25 \cdot \frac{[H_2SO_4]_0}{J_{\text{nuc}} + J_{\text{cond}}} \,,\, \Delta t_{\text{phys}} - \sum_{j=1}^{i-1} \Delta t_j \right) \tag{17}$$

where $J_{\text{nuc}}$ and $J_{\text{cond}}$ are updated at each timestep according to the updated value of $[H_2SO_4]$. The fourth and final sub-timestep is chosen so that the sum of all sub-timesteps is equal to one timestep of the model atmospheric physics $\Delta t_{\text{phys}}$.

$$\Delta t_4 = \max \left[ \min \left( 0.25 \cdot \frac{[H_2SO_4]_0}{J_{\text{nuc}} + J_{\text{cond}}}, \Delta t_{\text{phys}} - \sum_{j=1}^{3} \Delta t_j \right) \,,\, \Delta t_{\text{phys}} - \sum_{j=1}^{3} \Delta t_j \right] \tag{18}$$

This joint treatment of nucleation and condensation is imperfect but it has the advantage of being much more computationally efficient than usual solutions consisting of taking very short timesteps and much simpler than a simultaneous solving of nucleation and coagulation. The number of sub-timesteps could be increased for increased numerical accuracy, however a number of 4 sub-timesteps was considered to be sufficient. It should be noted that the processes of nucleation and condensation, as well as their competition, are only activated in the stratosphere.

### 2.2.6 Coagulation

The growing of sulfate particles through coagulation is represented through the semi-implicit, volume-conserving numerical scheme described in Jacobson et al. (1994). It is unconditionally stable even for timesteps of the order of hours. We restricted the coagulation kernel only to its main component, i.e. Brownian motion. Secondary components of coagulation due to convection, gravitation, turbulence or inter-particle van der Waals forces are neglected, which may partly explain the underestimation of particle size in Sect. 3. Sensitivity studies performed by English et al. (2013) and Sekiya et al. (2016) simulating the 1991 eruption of Mount Pinatubo found that including inter-particle van der Waals forces increased the peak effective radius by 10 % and decreased stratospheric AOD and burden by 10 %. Given that there are only few measurements on the van der Waals coagulation term, and the mixed results obtained in our model (see Sect. 3.3.1), we do not include this process in our default model, but offer it as an option in the code of the model (using the enhancement factors from Eq. (29) and (30) in Chan and Mozurkewich (2001)). Coagulation is only activated in the stratosphere.

For convenience, we repeat here the equations from Jacobson et al. (1994). New particles resulting from the coagulation of particles from size bins $i$ and $j$ have a combined particle volume $V_{i,j} = V_i + V_j$. They are distributed among the size bins according to the following definition of the distribution factor $f_{i,j,k}$:

$$
f_{i,j,k} =
\begin{cases}
\left( \dfrac{V_{k+1} - V_{i,j}}{V_{k+1} - V_k} \right) \dfrac{V_k}{V_{i,j}} & \text{for } V_k \leq V_{i,j} < V_{k+1} \ ; \ k < N_B \\[2ex]
1 - f_{i,j,k-1} & \text{for } V_{k-1} \leq V_{i,j} < V_k \ ; \ k > 1 \\[1ex]
1 & \text{for } V_{i,j} \geq V_k \ ; \ k = N_B \\[1ex]
0 & \text{otherwise}
\end{cases}
\tag{19}
$$

As discussed previously, the same distribution factor is applied for the other physical processes affecting particle size (i.e., nucleation, and the net effect from condensation and evaporation). To our knowledge, this is an original feature of our model. It should be noted that we have favoured conservation of aerosol mass (and volume) over conservation of aerosol number in all these processes.

The semi-implicit approach gives the following equation for the concentration of particles in bin $k$ after coagulation over a timestep $\Delta t$:

$$
V_k C_k^{(t+1)} = \frac{V_k C_k^{(t)} + \Delta t \sum\limits_{j=1}^{k} \left( \sum\limits_{i=1}^{k-1} f_{i,j,k}\, \beta_{i,j}\, V_i\, C_i^{(t+1)}\, C_j^{(t)} \right)}{1 + \Delta t \sum\limits_{j=1}^{N_B} (1 - f_{k,j,k})\, \beta_{k,j}\, C_j^{(t)}}
\tag{20}
$$

with $C_k^{(t)}$ the particle concentration in bin $k$ at timestep $t$, $C_k^{(t+1)}$ the particle concentration in bin $k$ at timestep $t+1$, and $\beta_{i,j}$ the coagulation kernel. For purely Brownian coagulation, the kernel has the form

$$
\beta_{i,j}^B = \frac{4\pi\,(r_i + r_j)(D_i + D_j)}{\dfrac{r_i + r_j}{r_i + r_j + (\delta_i^2 + \delta_j^2)^{1/2}} + \dfrac{4(D_i + D_j)}{(\bar{v}_{p,i}^2 + \bar{v}_{p,j}^2)(r_i + r_j)}}
\tag{21}
$$

with the particle diffusion coefficient

$$
D_i = \frac{k_B T}{6\pi r_i \eta} \left[ 1 + \text{Kn}_i \left( 1.249 + 0.42 \cdot \exp\left( \frac{-0.87}{\text{Kn}_i} \right) \right) \right]
\tag{22}
$$

where $\eta$ is the dynamic viscosity of air, the thermal velocity of a particle in bin $i$ with mass $m_i$

$$
\bar{v}_{p,i} = \sqrt{\frac{8 k_B T}{\pi m_i}}
\tag{23}
$$

the mean distance from the centre of a sphere reached by particles leaving the surface of the sphere and travelling a distance of the particle mean free path $\lambda_{p,i}$

$$
\delta_i = \frac{(2 r_i + \lambda_{p,i})^3 - (4 r_i^2 + \lambda_{p,i}^2)^{3/2}}{6 r_i \lambda_{p,i}} - 2 r_i
\tag{24}
$$

and the particle mean free path

$$\lambda_{\mathrm{p},i} = \frac{8\,D_i}{\pi\,\bar{v}_{\mathrm{p},i}} \tag{25}$$

### 2.2.7 Aerosol chemical composition and density

The weight fraction of $H_2SO_4$ in the aerosol as a function of temperature and $H_2O$ partial pressure is computed following the approach described in Steele and Hamill (1981) and also used in Tabazadeh et al. (1997). In this approach, the water content of the aerosol particles is assumed to be in equilibrium with the surrounding ambient water vapour. The composition is assumed to be constant over the whole particle size range.

The aerosol particle density as a function of temperature and $H_2SO_4$ weight fraction $w_{H_2SO_4}$ (in %) can then be computed from the rough approximation

$$\rho_p = \left(A \cdot w_{H_2SO_4}^2 + B \cdot w_{H_2SO_4} + C\right) \cdot \left(1 - \frac{0.02}{30}\,(T - 293)\right) \tag{26}$$

with the constants $A = 7.8681252 \cdot 10^{-6}$, $B = 8.2185978 \cdot 10^{-3}$, $C = 0.97968381$, and $T$ in K.

### 2.2.8 Sedimentation

Particles in the stratosphere sediment due to gravity with a velocity depending on their size and density and ambient pressure. The Stokes sedimentation velocity (with Cunningham correction) of a particle in size bin $k$ is given by:

$$v_{\mathrm{sed},k} = \frac{2\,\mathrm{g}\,r_k^2\,(\rho_{\mathrm{p}} - \rho_{\mathrm{air}})}{9\,\eta} \left[1 + \mathrm{Kn}_k \left(1.257 + 0.4 \cdot \exp\left(\frac{-1.1}{\mathrm{Kn}_k}\right)\right)\right] \tag{27}$$

with the gravity g, the particle density $\rho_{\mathrm{p}}$ and the air density $\rho_{\mathrm{air}}$.

The sedimentation process is computed with a semi-implicit scheme as described in Tompkins (2005). The concentration of particles in a bin $k$ (omitted here for clarity) in the model layer $j$ (with $j$ numbered from the top of the atmosphere to the surface) after sedimentation at timestep $t+1$ is given by:

$$C_j^{(t+1)} = \frac{C_j^{(t)} + C_{j-1}^{(t+1)} \cdot \dfrac{\rho_{j-1}\,v_{j-1}}{\rho_j\,\Delta z_j}\,\Delta t_{\mathrm{phys}}}{1 + \dfrac{\rho_j\,v_j}{\rho_j\,\Delta z_j}\,\Delta t_{\mathrm{phys}}} \tag{28}$$

with $v_j$ the sedimentation velocity, $\rho_j$ the air density and $\Delta z_j$ the thickness of layer $j$. The scheme is solved downwards, it is very stable and a timestep $\Delta t_{\mathrm{phys}}$ of 30 minutes is appropriate for our model vertical resolution. Unlike the aerosol processes described above, it is active not only above but also below the tropopause. It is applied to all bins of the aerosol size distribution but only has a noticeable impact on larger particle bins.

Once the particles cross the tropopause they are rapidly removed from the troposphere through wet and dry deposition. Parametrisations of dry and wet scavenging are those of the LMDZ model and are not described here as they have minimal impact on the aerosol stratospheric aerosol layer. They are nevertheless important to model the tropospheric fate and impact at the surface of aerosols or aerosol precursors injected in the stratosphere.

## 2.3 Aerosol optical properties

Averaged optical properties of the particles (extinction cross section $\sigma_i$ in $m^2$ per particle in bin $i$, asymmetry parameter $g_i$ and single scattering albedo $\omega_i$) are computed for each of the 6 SW and 16 LW spectral bands of the radiative transfer scheme using refractive index data from Hummel et al. (1988). We use our own Mie routine derived from Wiscombe (1979) and widely tested by the authors. In the SW, we account for variations of the incoming solar radiation within each band by computing aerosol optical properties at a higher spectral resolution (24 spectral bands) and weighting the properties with a typical solar spectrum. In the LW, we account for variations in the refractive index of the aerosols within each band by computing aerosol optical properties at a higher spectral resolution and weighting the properties with a black body emission spectrum using a typical stratospheric temperature of 220 K. To avoid Mie resonance peaks in the aerosol optical properties, we subdivide each aerosol size bin into 10 intervals which are logarithmically spaced and assume a uniform distribution within the size bin for computing average properties. For very small Mie parameters ($x < 0.001$), which occurs for the smallest particle bins and the longest wavelengths in the infrared, we extrapolate the Mie properties computed for $x = 0.001$ for numerical stability using known asymptotic behaviour of the scattering and absorption properties. Aerosol optical properties are computed once for each aerosol bin assuming a constant sulfuric acid mass mixing ratio of 75% and a temperature of 293 K (conditions for which the refractive index was measured) and are then integrated over the size distribution at every time step according to the actual local size distribution. Hence, the optical depth $\tau_k$, the single scattering albedo $\omega_k$ and the asymmetry parameter $g_k$ in model layer $k$ with particle concentrations $C_{i,k}$ (in particles per $m^3$) and the vertical extent $\Delta z_k$ (in m) can be computed as

$$\tau_k = \sum_{i=1}^{N_B} \sigma_i\, C_{i,k}\, \Delta z_k \qquad ; \qquad \omega_k = \frac{1}{\tau_k} \sum_{i=1}^{N_B} \omega_i\, \sigma_i\, C_{i,k}\, \Delta z_k \qquad ; \qquad g_k = \frac{1}{\tau_k\, \omega_k} \sum_{i=1}^{N_B} g_i\, \omega_i\, \sigma_i\, C_{i,k}\, \Delta z_k \qquad (29)$$

Aerosol optical properties are also computed at specific wavelengths (443, 550, 670, 765, 865, 1020 nm and 10 μm) for diagnostic purpose. It should be noted that in the LW, the RRTM model neglects scattering and only accounts for absorption. Hence we only feed the model with the LW absorption optical depth at each model layer.

## 3 Model validation

### 3.1 Non-volcanic background aerosol

The capability of our model to simulate a reasonable background stratospheric sulfate aerosol is tested by running the model for a decade with climatological OCS and $SO_2$ concentrations and lifetimes as the only boundary conditions. In this setup the model is not nudged to meteorological reanalysis.

The self-evolving aerosol distribution reaches a steady state or equilibrium (subject to seasonal variations) after about 5 years. In this steady state the global mean stratospheric aerosol optical depth (SAOD) at 550 nm is 0.002, which is in good agreement with the observed SAOD of 0.002-0.0025 at 525 nm (in the tropics and at mid-latitudes) during the period of very low stratospheric sulfur loading around the year 2000 (e.g., Vernier et al., 2011). The global stratospheric aerosol burden is

0.08 Tg S and the mean dry effective radius is 62 nm. The dry effective radius increases to 106 nm if only particles with radius larger than 50 nm, which make up 84 % of the burden, are taken into account.

Figure 4 shows that the aerosol layer is distributed over the whole globe, but is thicker and lower in altitude at high latitudes than in the tropics. The SAOD is highest at the summer pole. Unfortunately, there are too few observations and datasets (with a clear delineation of the tropopause) to validate or invalidate the latitudinal and seasonal distribution generated by our stratospheric aerosol model.

The comparison of the modelled particle size distribution at different latitudes shows that there are almost as many small particles at the Equator as at mid- and high-latitudes, but considerably less in the optically relevant size range. The aerosol layer is zonally quite homogeneous with deviations from the zonal mean value within $\pm 15$ % for optical depth and $\pm 25$ % for effective radius around $30°$ N/S and within $\pm 5$ % at low and high latitudes.

In Fig. 5 we compare the modelled size of the background aerosol to observations in May 2000, a period of very low stratospheric sulfate aerosol burden. While the modelled concentrations of particles with radius $r > 0.01\,\mu m$ and $r > 0.15\,\mu m$ in the lower stratosphere (below 19 km) match the observations quite well, the deviation increases with altitude. The concentration of larger particles with radius $r > 0.5\,\mu m$ is underestimated everywhere by the model by roughly one order of magnitude. This may be due to the fact that the observations are from a period still slightly influenced by precedent eruptions, i.e. not from pure background conditions. But the model is also missing secondary sources of stratospheric aerosol (e.g., meteoritic dust), which might be relevant in such a background case.

Figure 6 shows the modelled stratospheric sulfur budget under background conditions in steady state (11[th] year). Interestingly, the major part of the stratospheric $SO_2$ comes from the troposphere and only a minor part from the conversion of OCS occurring above the tropopause. This might be partly caused by the relatively long lifetime of OCS (here 8 years on average). The $SO_2$ is converted to $H_2SO_4$ with a lifetime of 36 days, while sulfuric acid has a lifetime of 44 days with respect to conversion into particles (considering both nucleation and condensation). The relatively short aerosol lifetime of 233 days can be explained by the fact that most of the aerosol is only slightly above the tropopause and at high latitudes, where it can enter the troposphere more easily and gets removed quickly via wet and dry deposition.

## 3.2 Mount Pinatubo eruption 1991

The eruption of Mount Pinatubo (Philippines) in June 1991 was the largest of the 20[th] century. Observations of the volcanic aerosols in the following months and years offer a unique opportunity to evaluate the performance of stratospheric aerosol models such as LMDZ-S3A under conditions of relatively high stratospheric sulfate loading.

In order to get a realistic spatial distribution of the aerosols in the simulation, horizontal winds are nudged to ECMWF ERA-interim reanalysis fields and sea surface temperatures (SST) are prescribed to their historical values. The simulation is initialised in January 1991 from the end of the 10 year spin up simulation (see Sect. 3.1). On June 15 1991, 7 Tg S in the form of $SO_2$ are injected into the grid cell including Mount Pinatubo at $15°$ N and $120°$ E over a period of 24 hours and vertically distributed as a Gaussian profile centred at 17 km altitude with a standard deviation of 1 km. This initial height was adjusted as a free parameter after comparing the resulting aerosol distribution of simulations with emission at 16, 17 and 18 km to

observations (see Fig. 7). This injection height may seem quite low compared to other simulations of the eruption, but it should be recalled that our model takes into account the interaction of aerosols with the radiation. The evolving aerosol has a net heating effect on the surrounding air through absorption of solar and terrestrial radiation (only partly compensated by emission of terrestrial radiation), which (together with the ascending branch of the Brewer-Dobson circulation) causes a significant uplift of the volcanic aerosol plume to more than 25 km altitude three months after the eruption. This radiatively driven uplift was already described by Aquila et al. (2012), who also found the best agreement between the simulated and the observed sulfate cloud if the $SO_2$ is injected at an altitude of 16 to 18 km.

The resulting spatial distribution of the aerosol extinction coefficient is compared to satellite and ground-based observations that are compiled in the CMIP6 aerosol data set [Luo Beiping, personal communication] in Fig. 7. The simulation with emission of $SO_2$ at 17 km was selected as the best fit, because the height of the maximum extinction coefficient at 1020 nm in September and December 1991 is closest to the CMIP6 data. In contrast, emission at 16 km results in faster meridional transport in the lower stratosphere and therefore a too fast decrease in aerosol extinction after the eruption, while emission at 18 km produces an aerosol layer considerably higher in altitude than observed.

The modelled evolution of the SAOD at 550 nm is also compared to a climatology from Sato (2012) and to SAOD simulated with the WACCM model by Mills et al. (2016) in Fig. 8. The global mean SAOD increases a little faster in LMDZ-S3A than in the Sato climatology, but just as fast as in WACCM. LMDZ-S3A slightly underestimates the maximum value of 0.15 from the Sato climatology and decreases at approximately the same rate of 7-8 % per month, while the decrease in WACCM, which includes several minor volcanic eruptions after Pinatubo, is slower. All three latitudinal distributions of the zonal mean SAOD are overall in good agreement, but with an earlier decrease in the tropics in LMDZ-S3A and with a stronger asymmetry towards the northern hemisphere in the WACCM simulation.

Figure 9 shows that the 7 Tg S emitted as $SO_2$ during the eruption are quickly converted into particles. The aerosol burden reaches its maximum 4 months after the eruption and then decreases slowly until it reaches a background value again after 4 to 5 years. The $H_2SO_4$ burden increases slower than the aerosol, probably because it requires more time to transport the sulfur to the higher stratosphere. This is the only region where a larger reservoir of sulfuric acid vapour can remain because particles tend to evaporate at the local temperature and pressure.

Particle size is compared to the continuous optical particle counter (OPC) measurements by Deshler et al. (2003) at 41° N in Fig. 10 and Fig. 11. The modelled stratospheric effective particle radius in the grid cell containing Laramie, Wyoming (41° N, 105° W) is a bit lower than measured by the OPC, but mostly within the given uncertainty of the measurement, if one takes into account particles of all sizes. However the sensitivity of the OPC to small particles with a radius below 0.15 μm (the smallest size class measured directly by the OPC) is not very well known. If only particles with a dry radius above 0.15 μm are considered and the smaller ones are completely ignored in the model, the effective radius is mostly overestimated by the model. But as the OPC's sensitivity to the small particles can be assumed to lie in between these two extreme cases (all or nothing), the agreement between modelled and observed particle size may be judged as good.

In Fig. 11 the modelled and the observed particle size distributions 5, 11 and 17 months after the eruption are compared. The model tends to overestimate particle concentrations of all size bins in the higher stratosphere, but reproduces the observations of

$r > 0.01\,\mu m$ and $r > 0.15\,\mu m$ particles fairly well at lower levels. The concentration of $r > 0.5\,\mu m$ particles is underestimated at the height of highest concentrations (17–21 km).

We also compare simulated and observed lower stratospheric (LS) temperature anomalies following the Pinatubo eruption, although this is far from being straightforward. A few studies report LS temperature changes following Mt. Pinatubo's eruption as measured from satellite with a microwave sounding unit (MSU) instrument (Randel et al., 1995; Zhang et al., 2013). As it relies on microwave, MSU has the advantage of not being influenced by the presence of aerosols, but it has the disadvantage of having a relatively broad weighting function on the vertical that encompasses both the upper troposphere (UT) and a large fraction of the stratosphere. Because stratospheric temperatures in climate models are generally biased one way or the other, it is not meaningful to compare absolute values of temperature. Rather we should compare temperature anomalies with and without the Pinatubo aerosols. There is a methodological issue here however. In the case of observations, one can only compare years 1991/1992 against a climatology from previous or following years. We could do the same in the model but lack a long enough simulation, so we compare two simulations with and without the Pinatubo aerosols and infer the temperature anomaly due to the aerosols.

Despite the intrinsic limitations of such a comparison, we compare here the model temperature anomaly (with and without Pinatubo aerosols) in nudged mode with the observed temperature anomaly (relative to baseline years) in Fig. 12. This is not equivalent because the stratospheric heating may have changed (and probably has changed) the stratospheric circulation, and therefore the temperature anomalies, which is not the case in the model. Zhang et al. (2013) report a peak warming of about 2 K after Pinatubo in the global mean for MSU channel 4 that encompasses both the upper troposphere and lower stratosphere. This is a bit more warming than shown in an earlier study by Randel et al. (1995). CMIP5 models show up to 3 K anomaly (see Fig. 2 in Zhang et al. (2013)). The temperature anomaly in the LMDZ-S3A simulation, convoluted with the MSU channel 4 weight function to make things comparable, is larger with a peak warming of 4.0 K, which may indicate an overestimated radiative heating in the model. One reason for the discrepancy in peak warming is likely due to the fact that $O_3$ is prescribed in the model to a constant climatology whereas, in the real world, $O_3$ has decreased by up to 15% after the Pinatubo eruption (according to Randel et al. (1995)). Since a large fraction of the SW heating rate induced by aerosols is actually related to an increase in gaseous absorption due to an increase in photon path upon aerosol scattering, the observed decrease in $O_3$ is expected to lead to a decrease in shortwave heating rates. We would need to couple an interactive $O_3$ scheme to our model to test this hypothesis. Finally it is also worth mentioning that the simulated temperature anomaly spreads from the tropics to the high latitudes within about a month (at least in nudged mode), that is more rapidly than the sulfate aerosol itself and the corresponding AOD in the visible spectral range. This behaviour is caused by a relatively strong diffusion of the temperature field in the model –which is required to stabilize the model dynamics– while the aerosol is transported through a less diffusive numerical scheme. The negative trend in the observed LS temperature anomaly in the years after the eruption is probably due to increasing concentrations of well-mixed greenhouse gases (GHG). The model does not show this behaviour because GHG concentrations do not vary in this experiment.

### 3.3 Sensitivity studies under Pinatubo conditions

#### 3.3.1 Sensitivity to van der Waals coagulation enhancement factor

In LMDZ-S3A we have only considered Brownian coagulation (Jacobson et al., 1994). Other terms for coagulation include those due to van der Waals forces, sedimentation and turbulence. Among these additional terms, only that due to van der Waals forces has been considered by some authors (English et al., 2013; Sekiya et al., 2016). Both studies rely on the calculations of Chan and Mozurkewich (2001), who measured coagulation for sulphuric acid particles of identical size end inferred an enhancement factor over Brownian coagulation for the limit cases of the diffusion (continuum) regime ($E(0)$) and the kinetic (free molecular) regime ($E(\infty)$). These enhancement factors are not directly usable in our model because stratospheric conditions encompass both the continuum and the free molecular cases and the equations in Jacobson et al. (1994) cover the general case. But in order to determine the impact of neglecting van der Waals forces, we applied the parametrizations of the enhancement factor of Chan and Mozurkewich (2001) to the coagulation kernels of Jacobson et al. (1994) and performed two additional simulations of the Pinatubo eruption: a first one with coagulation enhanced uniformly by the factor $E(0)$ and a second one with coagulation enhanced uniformly by the factor $E(\infty)$ (which is generally larger than $E(0)$). The actual enhancement factor for stratospheric conditions can be expected to lie in between these two cases.

As in previous studies, the van der Waals coagulation term improves the comparison to observation for particle number concentration (not shown) and particle average size (shown in Fig. 13), but it makes it a little worse for AOD as shown in Fig. 14, with the global-mean stratospheric AOD peaking too low (and too early) compared to the Sato climatology. Given that there are only few measurements on the van der Waals coagulation term, and the mixed results obtained in our model, we do not include this process in our default model, but offer it as an option in the code of the model.

#### 3.3.2 Sensitivity to the SO$_2$ chemical lifetime

A limitation of our model when simulating very large SO$_2$ injections might be the assumption of a constant SO$_2$ chemical lifetime (and hence a constant OH mixing ratio). Bekki (1995) showed that a constant SO$_2$ lifetime is not justified for an eruption as large as that of the Tambora. In order to test the sensitivity of our results to the assumed global SO$_2$ removal rate, we performed another Pinatubo simulation with SO$_2$ lifetimes increased by a factor 2 on the day of the eruption and decreasing linearly to normal values within one month. It appears unlikely that the OH effect impacted the global SO$_2$ lifetime beyond this factor 2, notably when compared with observational studies of the volcanic SO$_2$ decay. Analyses of SO$_2$ observations after the eruption give a global SO$_2$ lifetime ranging from 23 to 35 days (Bluth et al., 1992; Read et al., 1993). We find that the increase in assumed SO$_2$ lifetime delays and increases slightly the peak of the global-mean AOD (shown in Fig. 15). However overall the sensitivity to the SO$_2$ lifetime appears to be small. Therefore we conclude that using a prescribed chemical lifetime is probably not a major limitation of our model except for very large SO$_2$ injection rates, although it is desirable of course to improve the model in that respect in future studies.

## 4 Conclusions

In this article we have presented a newly developed sectional stratospheric sulfate aerosol (S3A) model as part of the LMDZ atmospheric general circulation model. A strength of our model is that it can readily be coupled to other components of the of the IPSL climate (and Earth system) model to perform climate studies. The S3A model includes a representation of sulfate particles with dry radii between $1\,nm$ and $3.3\,\mu m$ in currently 36 size bins, as well as the precursor gases OCS, $SO_2$ and $H_2SO_4$. The aerosol-relevant physical processes of nucleation, condensation, evaporation, coagulation and sedimentation are represented together with interactive aerosol optical properties and radiative transfer in 6 solar and 16 terrestrial spectral bands. The tropospheric fate of stratospheric sulfate aerosols is also simulated.

The comparison of model output and available observations for low and high sulfur loadings shows that LMDZ-S3A is an appropriate tool for studying stratospheric sulfate aerosols with a focus on the evolution of particle size distribution and the resulting radiative effects. Therefore it can be used for simulations of volcanic eruptions like that of Mount Pinatubo in 1991, or even larger ones like Tambora in 1815, for which studies with appropriate aerosol-climate models linking sulfur emission and climate impact derived from proxies are needed. It can also be used for simulations of deliberate stratospheric aerosol injections in order to study the efficacy and side effects of this proposed geoengineering technique.

Our model strength lies in the representation of aerosol microphysics with robust numerical schemes, but the model also has a few limitations. In particular it is simplified in terms of stratospheric chemistry and this will be the subject of future work as S3A can be coupled to the REPROBUS model which is also part of the IPSL Earth system model. Interactive ozone is also expected to help the model simulate a smaller heating rate and temperature anomaly in the presence of volcanic aerosols. Further developments will also include a more comprehensive treatment of the coagulation kernel and the possible interactions with other aerosol types (organics and meteoritic dust) in the stratosphere.

Finally it has to be stated that it has been a non-trivial task to gather observations of stratospheric aerosols for model evaluation. A fully validated, gridded stratospheric aerosol climatology in an easily usable format (like netCDF) with information on how gap filling is performed would tremendously facilitate the evaluation of model results. The stratospheric aerosol dataset produced for CMIP6 [Luo Beiping, personal communication] is a significant step in the right direction. Knowledge of (average) tropopause height would be particularly useful, so that vertically integrated quantities like AOD can be compared between model and observations and potential biases coming from differences in tropopause height can be detected. A more systematic reporting of observational uncertainties from both in situ and satellite data would be welcomed. This request extends to the provision of error covariances between measured quantities as these are needed to compute the error budget of combinations of observed quantities (as it is the case for the aerosol effective radius).

## 5 Code availability

The code of S3A can be downloaded along with the LMDZ model from http://lmdz.lmd.jussieu.fr. S3A code is mostly contained within a separate directory StratAer of the model physics and is activated at compilation with a CPP key. A model

configuration LMDZORSTRATAER_v6 containing the S3A module is also available within the modipsl/libIGCM model environment of the IPSL Earth System Model http://forge.ipsl.jussieu.fr/igcmg_doc.

*Author contributions.* C. Kleinschmitt developed most of the new parts of the model, performed the simulations, visualised and analysed the data, and wrote most of the article. O. Boucher had the original idea of the new model, assisted extensively in the development, evaluation and analysis, and contributed to many parts of the written article. S. Bekki contributed the basis of the model code for condensation, evaporation, and aerosol composition, and wrote parts of the sections on these processes and on the history of aerosol modelling. F. Lott developed and tuned the model's interactive QBO and wrote the section on this. U. Platt contributed to the discussions leading to and accompanying the model development.

*Competing interests.* The authors declare that they have no conflict of interest.

*Acknowledgements.* This article is a contribution to the DFG-funded priority program SPP 1689. We thank Luo Beiping for providing a compilation of stratospheric aerosol observation data and Josefine Ghattas for preparing the LMDZORSTRATAER_v6 configuration. The authors acknowledge computing time from the TGCC under the GENCI projects t2014012201, t2015012201 and t2016012201. The authors would like to thank two anonymous reviewers for their constructive comments and the Editor, Graham Mann, for his editorial review of the first submitted version of this manuscript which has led to significant improvements to the introduction section.

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

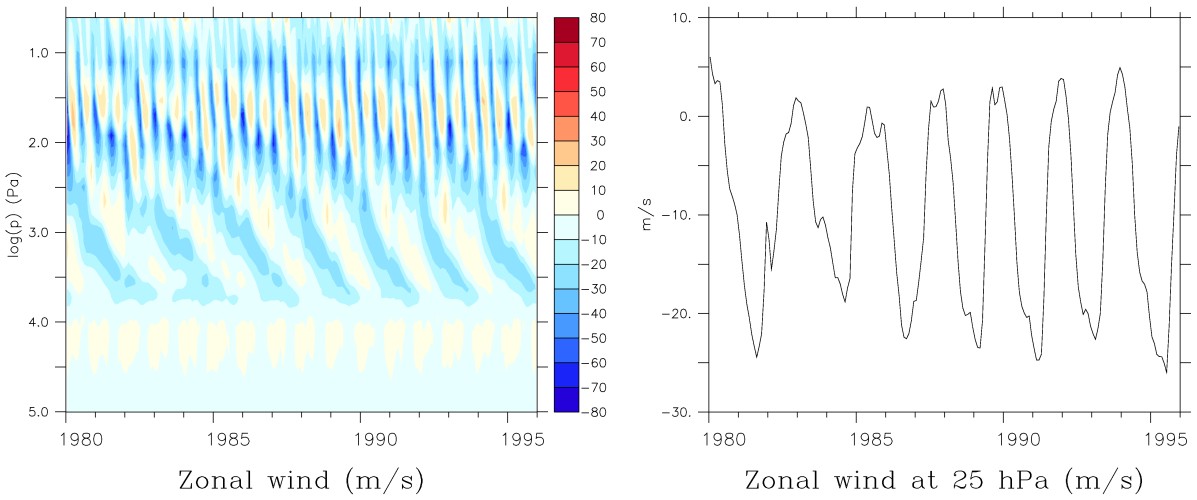

**Figure 1.** Left panel: Altitude-time profile of the zonal wind (in $m\,s^{-1}$), averaged between $10°$ S and $10°$ N, from a simulation with evolving background aerosol. The vertical axis shows the logarithm of the pressure in Pa. Right panel: Zonal wind at 25 hPa, averaged between $10°$ S and $10°$ N.

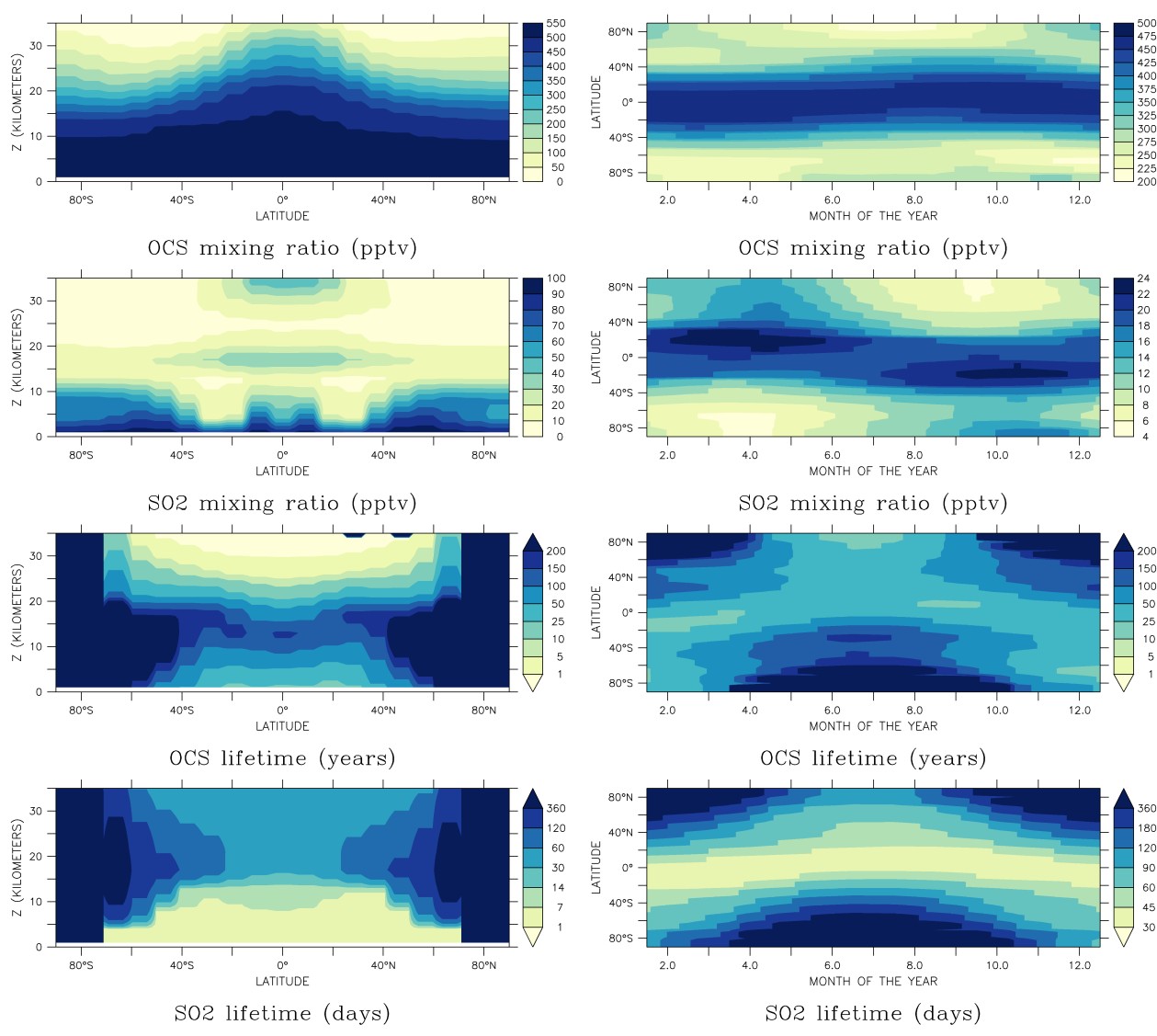

**Figure 2.** Climatological volume mixing ratios (upper half, in pptv) and lifetimes (lower half, in years or days) of OCS and SO$_2$ produced by the UPMC/Cambridge model and used as initial and boundary conditions for the LMDZ-S3A model. The left column shows the zonal and annual mean latitude-height distribution, while the right column shows an annual cycle of the zonal mean value at 20 km altitude.

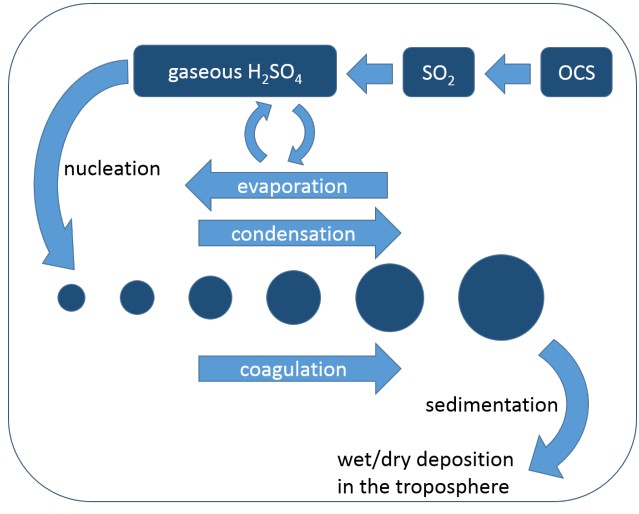

**Figure 3.** Schematic representation of the sulfur species and the processes affecting them that are represented in the LMDZ-S3A model.

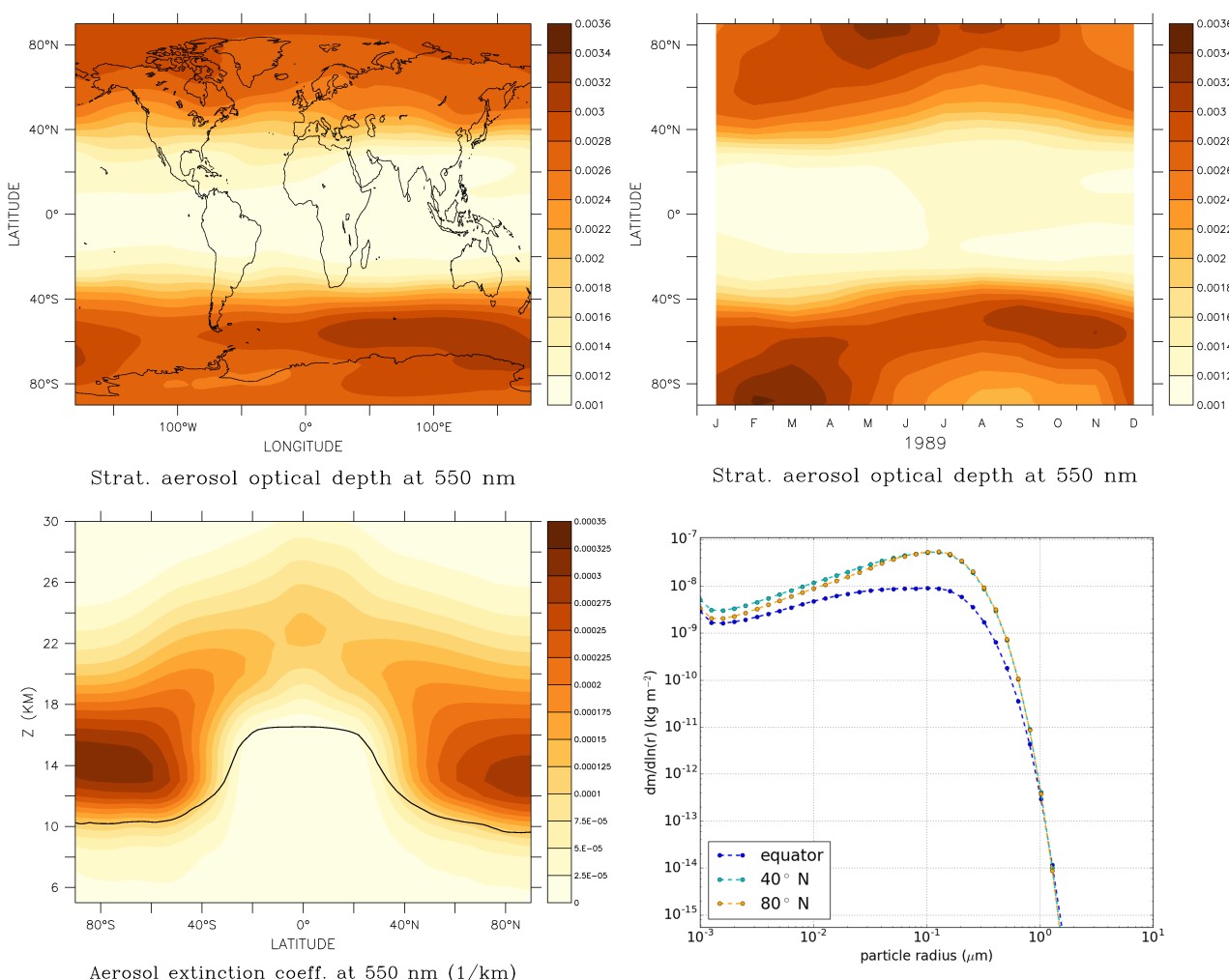

**Figure 4.** Upper left panel: Annual mean latitude-longitude distribution of the stratospheric AOD at 550 nm. Upper right panel: Zonal mean latitude-time distribution of the stratospheric AOD at 550 nm. Lower left panel: Zonal mean latitude-height distribution of stratospheric aerosol extinction coefficient ($km^{-1}$) at 550 nm. Lower right panel: Mass size distribution at different latitudes (in $kg\,m^{-2}$). All variables are from the $10^{th}$ year of the simulation with no volcanic input in the stratosphere and are assumed to represent a steady state.

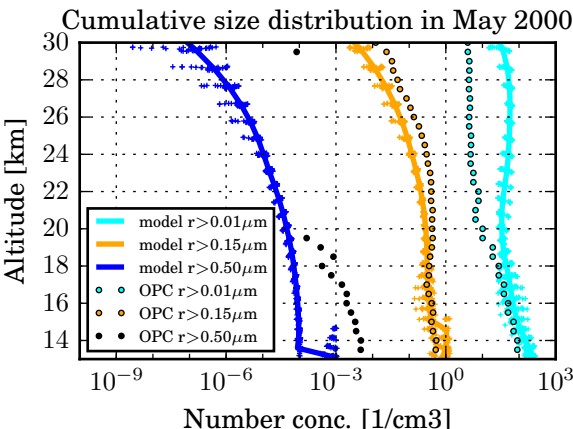

**Figure 5.** Vertical profile of the cumulative aerosol number concentration (cm$^{-3}$) for three channels ($r > 0.01\,\mu$m in light blue, $r > 0.15\,\mu$m in orange, and $r > 0.5\,\mu$m in dark blue) at Laramie, Wyoming (41° N, 105° W) in the style of Sekiya et al. (2016). Solid lines show the modelled monthly mean, while the crosses indicate the range of daily mean concentrations within that month. Optical particle counter (OPC) measurements from Deshler et al. (2003) are shown as symbols connected by dashed lines.

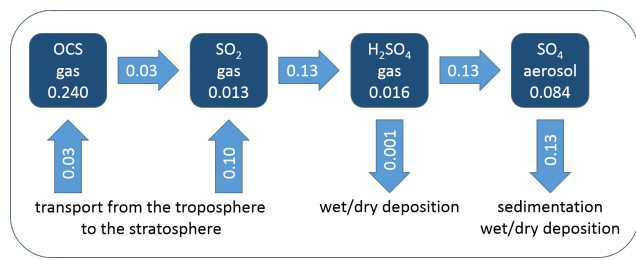

**Figure 6.** Modelled annual mean stratospheric burden (in Tg S) and fluxes (in Tg S yr$^{-1}$) of the represented sulfur species. The values are given for steady state background conditions without any stratospheric volcanic emissions. Advection can take the species out of the stratosphere into the troposphere, where they can be removed by wet and dry deposition.

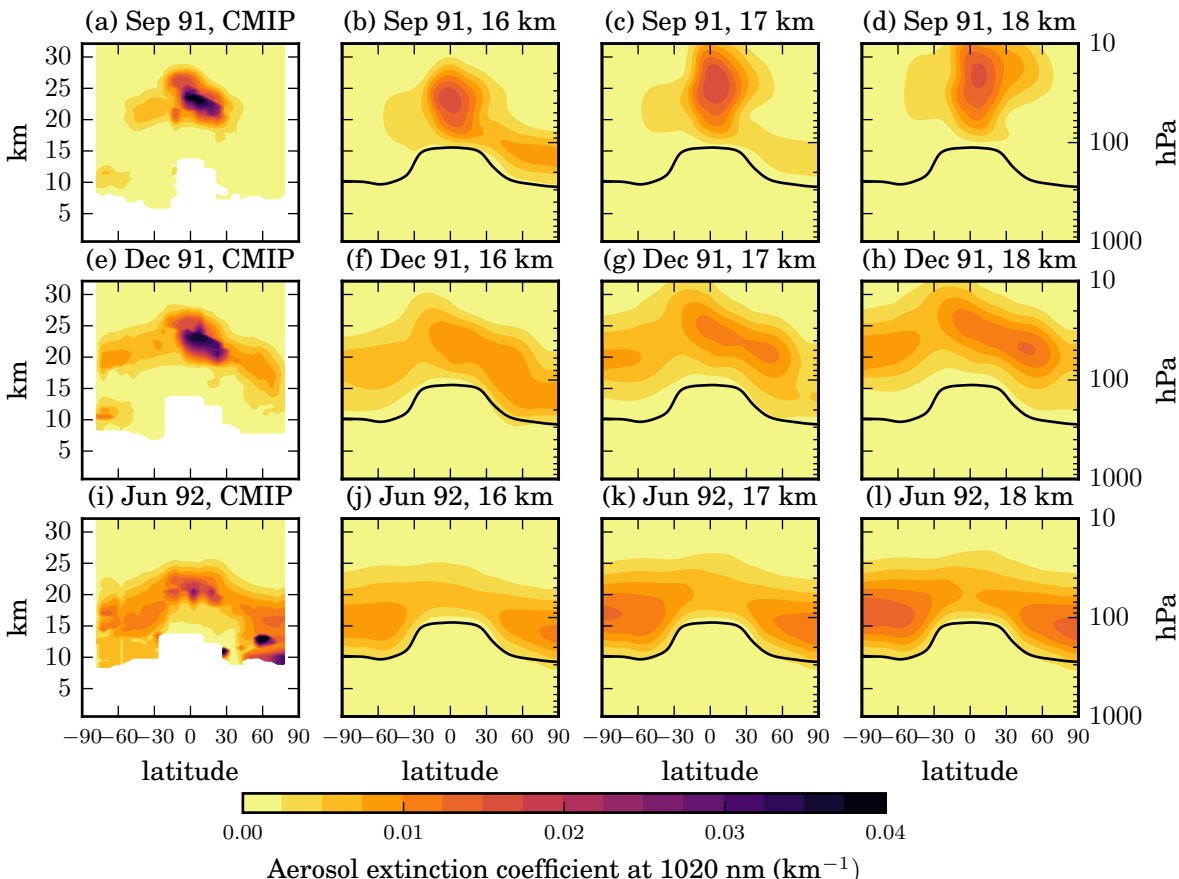

**Figure 7.** Temporal evolution of the zonal mean aerosol extinction coefficient ($km^{-1}$) at 1020 nm. Monthly mean latitude-height distributions in September 1991 (first row), December 1991 (second row) and June 1992 (third row). Observation-based CMIP6 aerosol data set (first column), simulations with emission of $SO_2$ at 16 km (second column), 17 km (third column) and 18 km (fourth column). The vertical axis shows the height in km and the black line indicates the modelled tropopause.

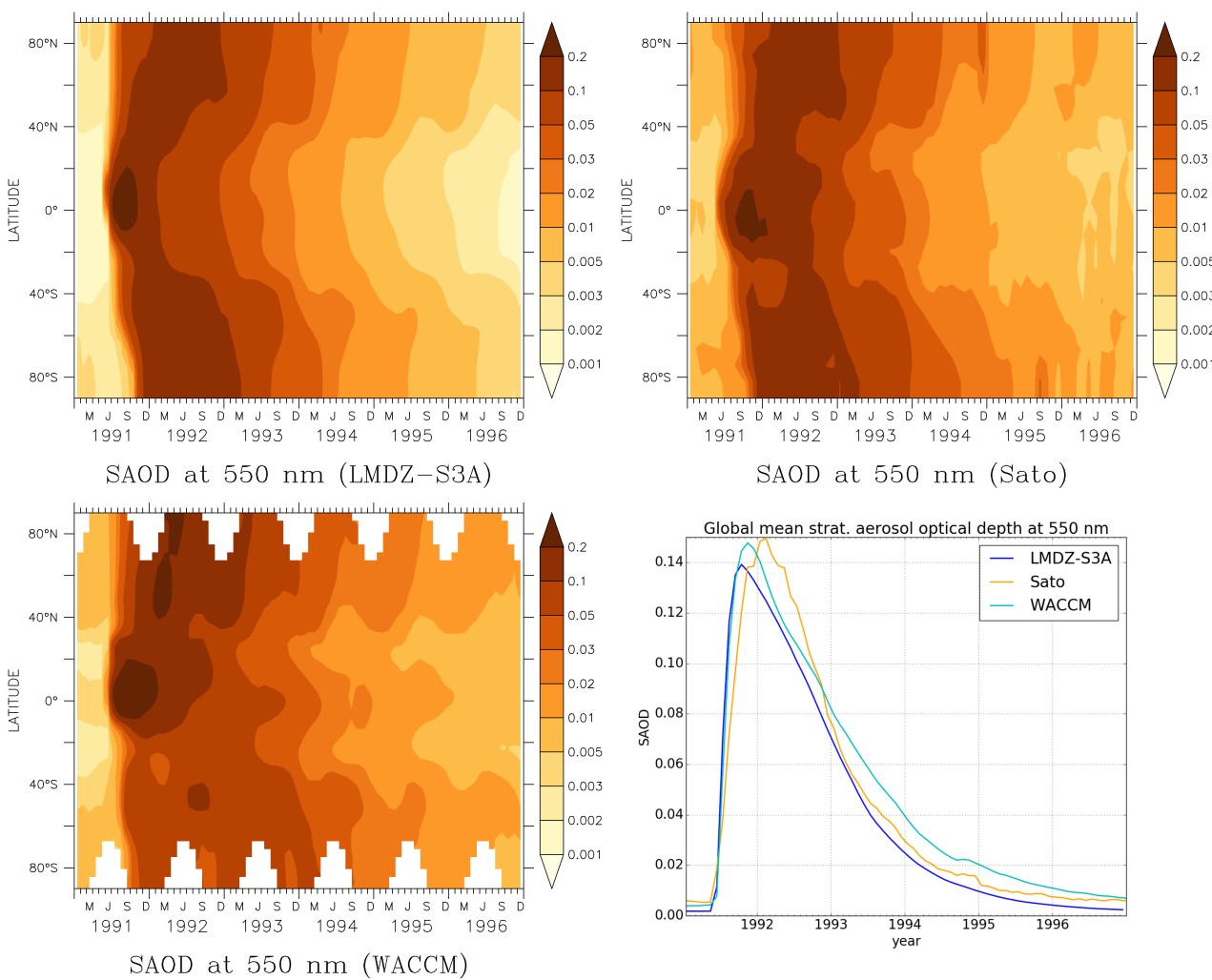

**Figure 8.** Evolution of the zonal mean stratospheric aerosol optical depth (SAOD) at 550 nm modelled with LMDZ-S3A (upper left panel) compared to the climatology from Sato (2012) (upper right panel) and to SAOD simulated with WACCM by Mills et al. (2016) (lower left panel), as well as the global mean SAOD (lower right panel). Note that unlike our simulation, WACCM includes small volcanic eruptions that occurred after that of Mount Pinatubo.

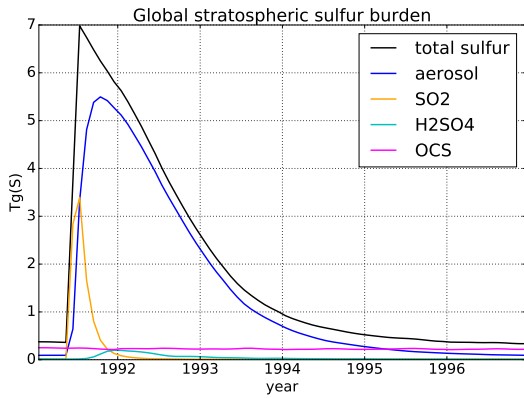

**Figure 9.** Evolution of the modelled stratospheric sulfur burden and its distribution among the different species for the period from January 1991 to December 1996, including the Pinatubo eruption (but no other eruptions).

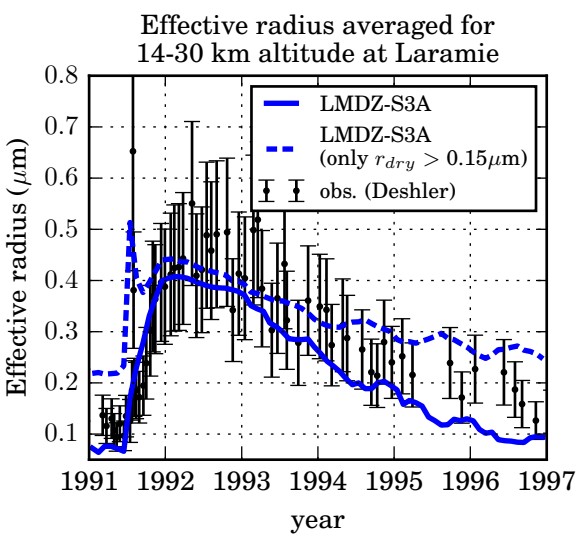

**Figure 10.** Stratospheric effective particle radius at Laramie, Wyoming (41° N, 105° W) as simulated by the LMDZ-S3A model and observed with optical particle counters (Deshler et al., 2003). Error bars of the measurements were determined from the 40 % uncertainty in aerosol surface area $A$ and volume $V$ assuming a correlation coefficient of 0.5 between $A$ at different altitudes, $V$ at different altitude and $A$ and $V$ at the same altitude.

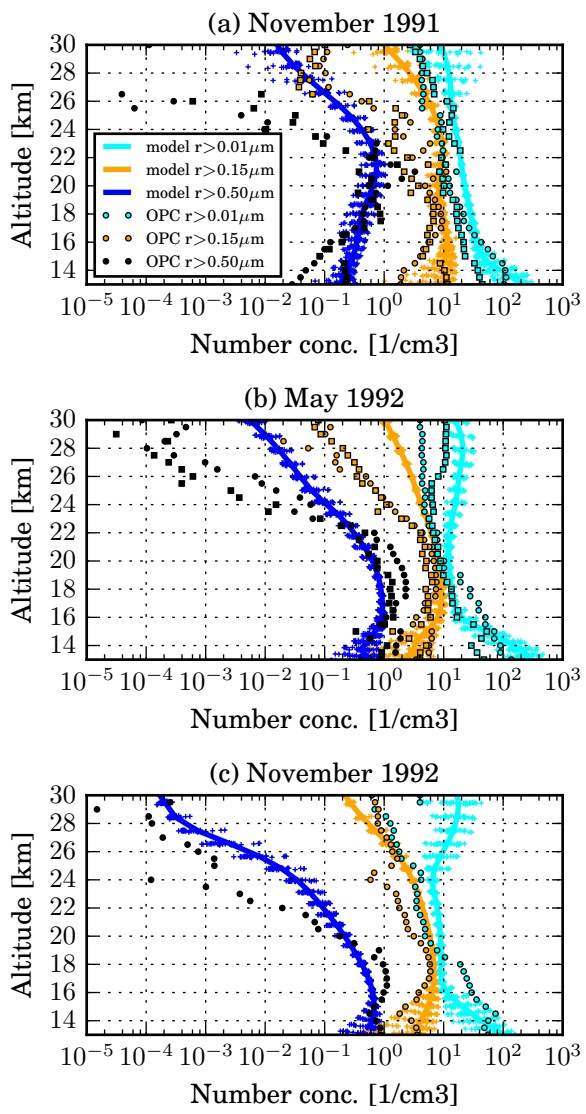

**Figure 11.** Vertical profile of the cumulative aerosol number concentration (cm$^{-3}$) for three channels ($r > 0.01\,\mu$m in light blue, $r > 0.15\,\mu$m in orange, and $r > 0.5\,\mu$m in dark blue) in November 1991, May 1992 and November 1992 at Laramie, Wyoming (41° N, 105° W) in the style of Sekiya et al. (2016). Solid lines show the modelled monthly mean, while the crosses indicate the range of daily mean concentrations within that month. Optical particle counter (OPC) measurements from Deshler et al. (2003) are shown as symbols.

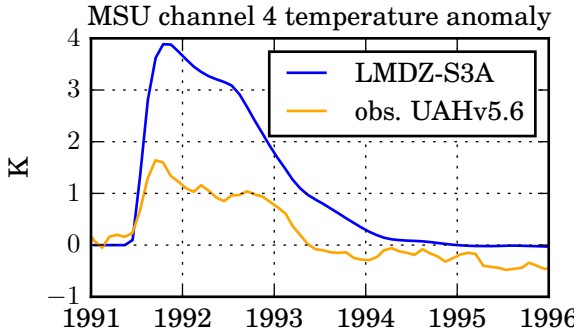

**Figure 12.** Global mean MSU channel 4 (upper tropospheric and lower stratospheric) temperature anomaly. The modelled anomaly due to the Pinatubo sulfate aerosol is computed as the difference between simulations with and without volcanic aerosol, while the observed anomaly reported by the UAH (Spencer and Christy, 1993) is with respect to the 1981–2010 base period.

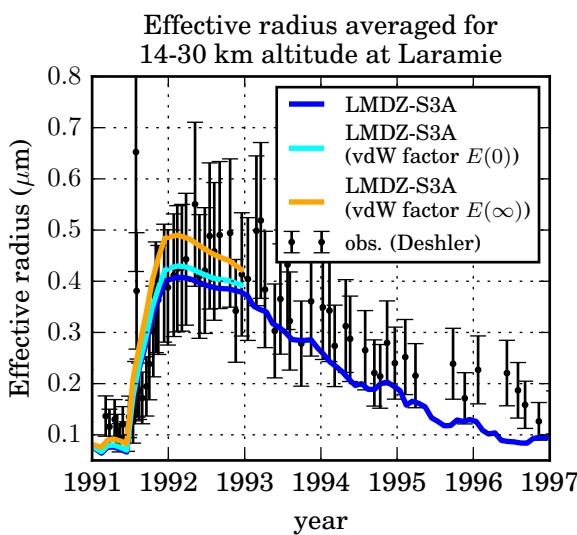

**Figure 13.** Stratospheric effective particle radius at Laramie, Wyoming (41° N, 105° W) as simulated by the LMDZ-S3A model and observed with optical particle counters (Deshler et al., 2003). The light blue (resp. orange) line shows the model result for coagulation enhanced by the continuum regime van der Waals enhancement factor $E(0)$ (resp. the kinetic regime enhancement factor $E(\infty)$).

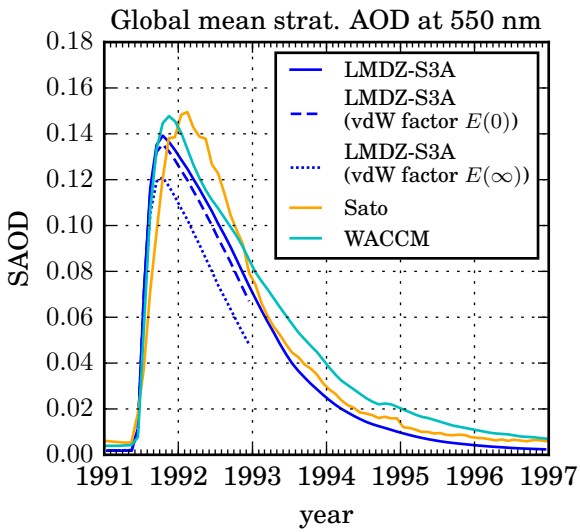

**Figure 14.** Evolution of the global mean stratospheric aerosol optical depth (SAOD) at 550 nm modelled with LMDZ-S3A compared to the climatology from Sato (2012) and to SAOD simulated with WACCM by Mills et al. (2016). The dashed (resp. dotted) line shows the model result for coagulation enhanced by the continuum regime van der Waals enhancement factor $E(0)$ (resp. the kinetic regime enhancement factor $E(\infty)$).

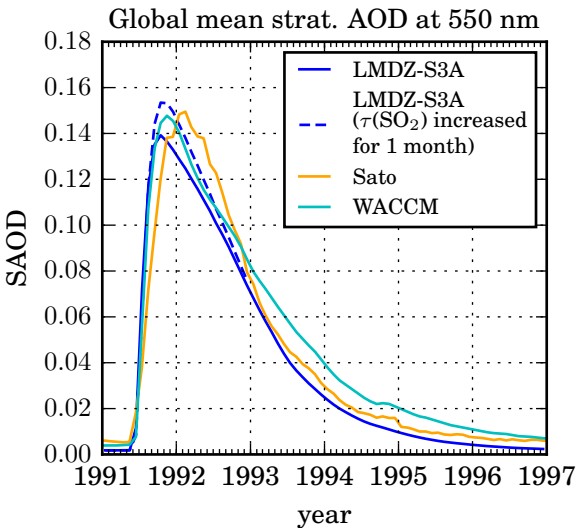

**Figure 15.** Same as Fig. 14, but here the dashed line shows the model result for an $SO_2$ lifetime doubled on the day of the eruption, decreasing linearly to climatological values within one month.