# Peer review of "The Sectional Stratospheric Sulfate Aerosol module S3A-v1 within the LMDZ general circulation model: Description and evaluation against stratospheric aerosol observations"

_Geoscientific Model Development, 2017_

## Referee Comment (RC1) · Anonymous Referee #1 · 13 May 2017

This paper describes a new microphysical aerosol module "S3A-v1" within the LMDZ GCM for stratospheric applications, and evaluates it against ambient and volcanic observations. This is valuable research, as stratospheric aerosols are important for climate and chemistry, and can have small (ambient) impacts or large perturbations such as volcanic eruptions or hypothetical geoengineering schemes. Additionally, as the authors note, due to the long lifetime, the growth and microphysical processes are complex and important, and most modeling studies lack at least one important process.

[Figure]

The model disclosed in this paper should be useful to advance our understanding of stratospheric aerosols.

This is a well-written paper with clear structure, meaningful utility, promising results, and good grammar. I have just a couple general suggestions and numerous minor specific suggestions to be considered before publication.

General Minor/Moderate suggestions:

1) Additional comparisons to observations can help strengthen the paper and understand how robust your model is. A few suggestions: a) Compare vertical profiles of sulfate aerosol mixing ratio taken by aircraft (Borrmann et al 2010; also applied in English et al. 2011 figure 10). b) Comparison to observations of sulfuric acid concentrations. (e.g. balloon data applied in English et al 2011 Figure 5c). c) Is there UTLS and stratospheric temperature data after Mt Pinatubo eruption (to test your aerosol radiative heating code)?

2) I suggest adding van der Waals forces to your coagulation scheme to improve simulations of ambient aerosol. In your Figure 5, small particles are overestimated and large particles are underestimated by about an order of magnitude in the middle/upper troposphere. Inclusion of van der Waals forces would significantly reduce this bias. This was investigated in the WACCM/CARMA model (English et al. 2011). WACCM/CARMA with van der Waals forces included in the coagulation scheme had a much better match to aerosol size distribution than your model (Fig 9 in English et al. 2011). This was concluded to be due to van der Waals forces; the experiment without van der Waals forces overestimated particle number (Fig 10 in English et al 2011), similar to your model. (Neither model includes meteorites, suggesting that not including meteorites is not the problem). This should improve your comparisons in your Figure 5 and 11, and would be a nice improvement to your model. Also, you could compare vertical profiles of mixing ratio (Borrmann et al 2010). Also, you could compare Pinatubo simulations to other studies that looked at van der Waals forces (English et al. 2013, Sekiya et al. 2016), although as you mention, the observed variability is too large to conclude with confidence whether including van der Waals forces improves volcanic simulations. Most likely, van der Waals forces is important to match ambient observations of number concentration though.

3. Using prescribed chemical lifetimes is probably fine for ambient aerosol, but could be problematic with large volcanic eruptions or geoengineering. I suggest you clarify these caveats and either do some evaluations that conclude that prescribing the oxidants doesn't cause too many errors in your results, or state that this current model setup is not recommended for very large volcanic eruptions or geoengineering. Also, cite Bekki 1995 for evidence of limited oxidants. (It's reassuring that your results are reasonable for Pinatubo, but unclear whether your model is safe to use for eruptions larger than Pinatubo or for geoengineering)

Specific suggestions:

Abstract: 1) Add that stratospheric aerosols impact chemistry as well as radiative forcing 2) Describe your model in more detail (it computes aerosol radiative heating and QBO but has semi-prescribed chemistry). 3) Remove the word "all" from relevant microphysical processes, as certainly *some* are missing, such as van der Waals forces. 4) Add a few more details of your results, e.g. number of ambient large particles are underestimated/small particles overestimated (which may be due to not including van der Waals forces)

Introduction Line 21: Remove the word "quite"

2.1.1 Line 20: Does Hourdin et al 2006 evaluate the model for stratospheric applications? If not, I suggest adding a subsection where you describe and evaluate some of the stratospheric processes in your model to ensure it is behaving reasonably, such as looking at strat age-of-air, trop-strat transport processes, seasonal/latitudinal temperatures. It is nice that you looked at QBO.

[Figure]

2.1.2 Line 8: Say why you only compute nucleation, condensation etc in the stratosphere (to save computational time? or is a different aerosol module used in the troposphere?)

2.1.2 Line 10: Do you compute the tropopause level at each timestep? Please clarify.

2.2.1 Line 19: I suggest put these paragraphs in a new section called "Chemistry".

2.2.1 Line 31: Please add some discussion regarding my general concern #3 here.

2.2.2 Line 5: The statement that coagulation and growth, not nucleation, determines particle size distribution is a main conclusion from English et al. 2011.

2.2.5 Line 13: Please add discussion of the possible importance of van der Waals forces to accurately model ambient aerosol. Results by English et al. 2011 (see Figure 9) suggest van der Waals forces is very important to get ambient aerosol correct (likely more important for ambient aerosol than for large perturbations such as volcanic eruptions).

2.3 Line 23: "widely tested by the authors". are there any peer-reviewed papers to cite?

3.1 Line 29+: (discuss the likely importance of van der Waals forces). Also, since English et al. 2011 also didn't have meteoritic dust in their model, but the size distribution looks good, the lack of meteoritic dust is not likely a source of error for particle size/mixing ratio (but probably is a source of error for burden/extinction).

3.2 Line 21: Cite Aquila et al 2012 who also investigated the relationship between injection height and Pinatubo accuracy when aerosol heating is involved (How do your injection height results compare )

3.2 Line 26: Is this data published anywhere yet?

Conclusions Line 30: Please add discussion about whether your model is safe to use for perturbations larger than Pinatubo and/or for hypothetical geoengineering schemes

given that oxidants are prescribed.

Figure 5 and Figure 11: It is difficult to distinguish the model and observation lines, especially the dark blue lines. Can you change the colors or markers to better distinguish?

Figure 7: Add to the caption the source of the CMIP6 aerosol data set. Please add a header to each row and column describing each panel.

References

Aquila, V., L. D. Oman, R. S. Stolarski, P. R. Colarco, and P. A. Newman (2012), Dispersion of the volcanic sulfate cloud from a Mount Pinatubo–like eruption, J. Geophys. Res., 117, D06216, doi:10.1029/2011JD016968.

Bekki, S. (1995), Oxidation of volcanic SO2: A sink for stratospheric OH and H2O, Geophys. Res. Lett., 22(8), 913–916, doi:10.1029/95GL00534.

Borrmann, S., Kunkel, D., Weigel, R., Minikin, A., Deshler, T., Wilson, J. C., Curtius, J., Volk, C. M., Homan, C. D., Ulanovsky, A., Ravegnani, F., Viciani, S., Shur, G. N., Belyaev, G. V., Law, K. S., and Cairo, F.: Aerosols in the tropical and subtropical UT/LS: in-situ measurements of submicron particle abundance and volatility, Atmos. Chem. Phys., 10, 5573–5592, doi:10.5194/acp-10-5573-2010, 2010.

Sekiya, T., K. Sudo, and T. Nagai (2016), Evolution of stratospheric sulfate aerosol from the 1991 Pinatubo eruption: Roles of aerosol microphysical processes, J. Geophys. Res. Atmos., 121, 2911–2938, doi:10.1002/2015JD024313.

---

## Referee Comment (RC2) · Anonymous Referee #2 · 26 May 2017

This manuscript describes the stratospheric aerosol model S3A and it's application in the LMDZ GCM. The model allows for interaction between aerosol radiative effects and atmospheric dynamics. Comparisons with observation for a background (unperturbed) period and for the Pinatubo period are presented. The introductory section has a thorough discussion of the evolution of global aerosol models and trade-offs related to the aerosol approach chosen. The sectional approach to aerosol is appropriate for the stratosphere. The disconnect between stratospheric and tropospheric aerosols should

be justified or it's limitations discussed and relegated to future model development. In particular, interactions between sulfate and organic aerosols in the UTLS region (see recent GRL paper by Yu et al., doi:10.1002/2016GL 070153) and affects of descending sulfate aerosol on clouds and tropospheric chemistry should be mentioned. Meteoritic particles in the middle stratosphere and polar regions could also be significant in some cases. The radiative code is appropriate to the application, with 6 shortwave and 16 longwave bands. The paper is generally well-written, though a few relevant model details and caveats have been omitted, as detailed below. The model has room to grow by adding an interactive chemical scheme and strat-trop aerosol interactions, but the version presented here is still a useful contribution to the literature and worthy of documentation in Geoscientific Model Development.

Specific Scientific Comments: Page 1, lines 16-18: "Gravitational sedimentation . . . is extremely dependent on the size of the aerosol particles and ambient air density." Air density (or pressure) explains why sedimentation is not important in the troposphere but is in the stratosphere.

Page 4, lines 6-7: It is not accurate to say that "We have included processes relevant to . . . much larger and/or longer emission rates than experienced in typical volcanic eruptions" because of the prescribed oxidants converting SO2 and OCS to sulfate. Perturbations to OH will decrease SO2 lifetime following an eruption much larger than Pinatubo. The model currently as the hooks to account for larger eruptions in the future when the REPROBUS chemical scheme is integrated into LMDZ, as explained on page 8, but currently does not have that capability.

Page 4, lines 23-25: Please include the height of the model top.

Page 7, lines 24-26: Apparently the photochemical transformation of H2SO4 gas to SO3 and SO2 above the top of the aerosol layer is neglected? This will result in errors in nucleation in the polar regions due to downwelling of SO2.

Page 8: lines 7-8: Wet and dry deposition are mentioned but not washout/cloud removal processes. Are aerosols removed in clouds throughout the troposphere or at the surface only?

Page 12, lines 17-24: Can you justify using the 1981 Steele and Hamill formulation for aerosol weight percent? How does it compare to Tabazadeh et al. 1997?

Page 13, lines 24: How can you justify using a constant temperature and 75 weight percent aerosol in the optical properties calculation for all latitudes and altitudes? This weight percent will be pretty far off near the poles and near the top of the aerosol layer. How much error does this contribute to the scattering and heating rates calculations?

Page 15-16, Pinatubo Experiment: It would be nice to see a figure showing the change in temperature, particularly near the tropical tropopause, due to the volcanic aerosols, and possibly a figure showing the change in stratospheric dynamics.

Minor comments: Page 2, line 30: "…a dozen global three-dimensional stratospheric aerosol models". Omit "of"

Page4, line 1: "Recent reviews of scientific studies…" Change "on" to "of"

Page 6, line 20: "Nudging is activated in the model calculations described in Sect 3.2"

---

## Author Comment (AC1) · 17 Jul 2017

**Final author response to the Referees' comments on "The Sectional Stratospheric Sulfate Aerosol module S3A-v1 within the LMDZ general circulation model: Description and evaluation against stratospheric aerosol observations"**

Christoph Kleinschmitt[1], Olivier Boucher[2], Slimane Bekki[3], François Lott[4], and Ulrich Platt[1]

[1]Institute of Environmental Physics, Heidelberg University, Im Neuenheimer Feld 229, 69120 Heidelberg, Germany
[2]Institut Pierre-Simon Laplace, CNRS / UPMC, 4 Place Jussieu, 75252 Paris Cedex 05, France
[3]Laboratoire Atmosphères Milieux Observations Spatiales, Institut Pierre-Simon Laplace, CNRS / UVSQ, 11 Boulevard d'Alembert, 78280 Guyancourt, France
[4]Laboratoire de Météorologie Dynamique, Institut Pierre-Simon Laplace, CNRS / ENS, 24 Rue Lhomond, 75231 Paris Cedex 05, France

*Correspondence to:* Christoph Kleinschmitt (christoph.kleinschmitt@iup.uni-heidelberg.de)

We thank both Referees for their thorough evaluation of the manuscript. Their comments are repeated below in blue and our response follows in black.

**Response to comments by Referee 1**

This paper describes a new microphysical aerosol module "S3A-v1" within the LMDZ GCM for stratospheric applications, and evaluates it against ambient and volcanic observations. This is valuable research, as stratospheric aerosols are important for climate and chemistry, and can have small (ambient) impacts or large perturbations such as volcanic eruptions or hypothetical geoengineering schemes. Additionally, as the authors note, due to the long lifetime, the growth and microphysical processes are complex and important, and most modeling studies lack at least one important process. The model disclosed in this paper should be useful to advance our understanding of stratospheric aerosols. This is a well-written paper with clear structure, meaningful utility, promising results, and good grammar. I have just a couple general suggestions and numerous minor specific suggestions to be considered before publication.

We thank the Referee for their general positive appreciation.

General Minor/Moderate suggestions:

1) Additional comparisons to observations can help strengthen the paper and understand how robust your model is. A few suggestions: a) Compare vertical profiles of sulfate aerosol mixing ratio taken by aircraft (Borrmann et al. (2010); also applied in English et al. (2011) figure 10). b) Comparison to observations of sulfuric acid concentrations. (e.g. balloon data applied in English et al. (2011) Figure 5c). c) Is there UTLS and stratospheric temperature data after Mt Pinatubo eruption (to test your aerosol radiative heating code)?

The manuscript already includes a number of comparisons to observations, but we agree that comparison to additional observations can bring further insight into the model. We consider below the three suggestions made by the Referee.

a) In Fig. 1 below we compare the simulated sulfate particle mixing ratio in the tropical upper troposphere / lower stratosphere (UT/LS) to observations from aircraft reported by Borrmann et al. (2010). As already seen on Fig. 5 in the discussion paper, the comparison to optical particles counter (OPC) measurements by Deshler et al. (2003) shows that the model overestimates the number of particles significantly, mainly at the lower end of the size distribution. This positive bias is larger at higher altitudes. Said differently, the height of the maximum particle mixing ratio is shifted towards higher potential temperatures (Fig. 1). There could be several reasons for these discrepancies: the model may predict too many small particles (but sensitivity tests described below on enhancing coagulation are not entirely conclusive), the model may predict the right amount of particles but at a too high altitude, or measurements may miss a fraction of small particles. The discrepancy can also be explained by a combination of these reasons.

[Figure]

**Figure 1.** Vertical profile of sulfate particle mixing ratios measured from aircraft in the tropics under background conditions (Borrmann et al., 2010) compared to simulated profiles at the same latitudes. The measurement campaigns TROCCINOX (TR), SCOUT-O3 (SC), and SCOUT-AMMA (AM) took place in 2005 and 2006 at various tropical sites (21°S, 12°S, and 12°N, respectively). The authors would like to thank Ralf Weigel for providing them with the measurement data.

b) As it was not possible to get access to the original measurement data from Arnold et al. which are shown in Fig. 5c of English et al. (2011), we performed an eyeball comparison between English's plot and the vertical profile of $H_2SO_4$ number concentration at 43°N in September and October in LMDZ-S3A under background conditions shown in Fig. 2. The agreement between the simulated and the measured sulfuric acid concentration is very good. In both cases the highest concentrations are found at an altitude of 30–35 km and reach values of $10^7$ to $10^8$ molecules per $cm^3$, while concentrations decrease to values

of $10^5$ to $10^6$ in the lower stratosphere. Only above $\approx 40$ km our model overestimates concentrations considerably, because it does not include photodissociation of $H_2SO_4$, which causes the concentration to decrease more rapidly with increasing altitude in the observations.

[Figure]

**Figure 2.** Simulated vertical profile of $H_2SO_4$ number concentration at $43°N$ in September and October under background conditions. To be compared to measurements by Arnold et al. shown in Fig. 5c of English et al. (2011).

c) Comparing simulated and observed temperature anomalies following the Pinatubo eruption is far from being straightfor-
ward. Upper tropospheric and lower stratospheric (UT/LS) temperatures are routinely measured by radiosondes but require some complicated processing and bias correction that are beyond the scope of this study. Temperatures are also retrieved from radiance measurements made by satellite infrared sounders such as AIRS or IASI, but the presence of aerosols complicates significantly the temperature retrieval. We are not aware of a reliable dataset of LS temperatures from infrared sounding during Pinatubo. Available meteorological reanalyses (such as ERA-interim) are not so reliable either because the meteorological
models used for deriving the reanalysis generally do not consider the effects of stratospheric aerosols. Hence the resulting re-analysis of the LS temperature is likely to be biased low. A few studies report LS temperature changes following Mt. Pinatubo's eruption as measured from satellite with a microwave sounding unit (MSU) instrument (Randel et al., 1995; Zhang et al., 2013). As it relies on microwave, MSU has the advantage of not being influenced by the presence of aerosols, but it has the disad-vantage of having a relatively broad weighting function on the vertical that encompasses both the UT and a large fraction
of the stratosphere. Because stratospheric temperatures in climate models are generally biased one way or the other, it is not meaningful to compare absolute values of temperature. Rather we should compare temperature anomalies with and without the Pinatubo aerosols. There is a methodological issue here however. In the case of observations, one can only compare years 1991/1992 against a climatology from previous or following years. In the case of the model, it is also possible to perform two simulations with and without the Pinatubo aerosols and infer the temperature anomaly due to the aerosols. This can be done in

nudged mode (which is the case of the simulations performed in this manuscript) or in free-running mode. In order to be able to compare the models results to the aerosol profiles measured at Laramie after the Pinatubo eruption, the model dynamics, here horizontal winds, has to be nudged towards meteorological analyses. Indeed, in a free-running mode, the GCM calculates its own "random" wind fields and cannot reproduce the actual dynamics and variability for specific years over a specific site such as Laramie. The wind nudging is possibly interfering with modelled temperature response, especially at high latitudes where dynamical feedbacks are known to play an important role.

Despite the intrinsic limitations of such a comparison, we compare here the model temperature anomaly (with and without Pinatubo aerosols) with the observed temperature anomaly (relative to baseline years) in Fig. 3. Zhang et al. (2013) report a peak warming of about 2 K after Pinatubo in the global mean for MSU channel 4 that encompasses both the upper troposphere and lower stratosphere. This is a bit more warming than shown in an earlier study by Randel et al. (1995). CMIP5 models show up to 3 K anomaly (see Fig. 2 in Zhang et al. (2013)). The temperature anomaly in the LMDZ-S3A simulation, convoluted with the MSU channel 4 weight function to make things comparable, is larger with a peak warming of 4.0 K, which may indicate an overestimated radiative heating in the model. One reason for the discrepancy in peak warming is likely due to the fact that $O_3$ is prescribed in the model to a constant climatology whereas, in the real world, $O_3$ has decreased by up to 15% after the Pinatubo eruption (according to Randel et al. (1995)). Since a large fraction of the SW heating rate induced by aerosols is actually related to an increase in gaseous absorption due to an increase in photon path upon aerosol scattering, the observed decrease in $O_3$ is expected to lead to a decrease in shortwave heating rates. We would need to couple an interactive $O_3$ scheme to our model to test this hypothesis. Finally it is also worth mentioning that the simulated temperature anomaly spreads from the tropics to the high latitudes within about a month (at least in nudged mode), that is more rapidly than the sulfate aerosol itself and the corresponding AOD in the visible spectral range. This behaviour is caused by a relatively strong diffusion of the temperature field in the model –which is required to stabilize the model dynamics– while the aerosol is transported through a less diffusive numerical scheme.

[Figure]

**Figure 3.** Global mean MSU channel 4 (upper tropospheric and lower stratospheric) temperature anomaly. The modelled anomaly due to the Pinatubo sulfate aerosol is computed as the difference between simulations with and without volcanic aerosol, while the observed anomaly reported by the UAH (Spencer and Christy, 1993) is with respect to the 1981–2010 base period.

2) I suggest adding van der Waals forces to your coagulation scheme to improve simulations of ambient aerosol. In your Figure 5, small particles are overestimated and large particles are underestimated by about an order of magnitude in the middle/upper troposphere. Inclusion of van der Waals forces would significantly reduce this bias. This was investigated in the WACCM/CARMA model (English et al., 2011). WACCM/CARMA with van der Waals forces included in the coagulation scheme had a much better match to aerosol size distribution than your model (Fig 9 in English et al. (2011)). This was concluded to be due to van der Waals forces; the experiment without van der Waals forces overestimated particle number (Fig 10 in English et al. (2011)), similar to your model. (Neither model includes meteorites, suggesting that not including meteorites is not the problem). This should improve your comparisons in your Figure 5 and 11, and would be a nice improvement to your model. Also, you could compare vertical profiles of mixing ratio (Borrmann et al., 2010). Also, you could compare Pinatubo simulations to other studies that looked at van der Waals forces (English et al., 2013; Sekiya et al., 2016), although as you mention, the observed variability is too large to conclude with confidence whether including van der Waals forces improves volcanic simulations. Most likely, van der Waals forces is important to match ambient observations of number concentration though.

In the original manuscript we have only considered Brownian coagulation for spherical neutral spheres (Jacobson et al., 1994). Other contributions for coagulation come from van der Waals forces, sedimentation and turbulence. Among these additional terms, only that due to van der Waals forces has been considered by some authors (English et al., 2013; Sekiya et al., 2016). Both studies rely on the calculations of Chan and Mozurkewich (2001), but do not describe precisely how this additional process is treated in their models. Chan and Mozurkewich (2001) measured coagulation for sulphuric acid particles of identical size. They infer an enhancement factor over Brownian coagulation for the limit cases of the diffusion (continuum) regime ($E(0)$) and the kinetic (free molecular) regime ($E(\infty)$). These enhancement factors are not directly usable in our model because stratospheric conditions encompass both the continuum and the free molecular cases and the equations in Jacobson et al. (1994) cover the general case. In additional sensitivity tests, we have now applied the parametrizations of the enhancement factor of Chan and Mozurkewich (2001) to the coagulation kernels of Jacobson et al. (1994). We performed two additional simulations of the Pinatubo eruption, a first one with coagulation enhanced uniformly by the factor $E(0)$ and a second one with coagulation enhanced uniformly by the factor $E(\infty)$, which is generally larger than $E(0)$. Actual enhancement factors corresponding to stratospheric conditions can be expected to lie in between these two limit cases. As anticipated by the Referee, the van der Waals coagulation term improves the comparison to observation for particle number concentration (Fig. 4) and effective radius (Fig. 5), but it makes it a little worse for AOD as shown in Fig. 6, with global-mean AOD peaking too low (and too early) compared to the Sato climatology. Given that there are only few measurements of the van der Waals coagulation term, and the mixed results obtained in our model, we do not include this process in the default version of our model, but we offer it as an option in the revised code. The manuscript has been revised to reflect this discussion.

[Figure]

**Figure 4.** Vertical profile of the cumulative aerosol number concentration (cm$^{-3}$) for three channels ($r > 0.01\,\mu$m in light blue, $r > 0.15\,\mu$m in orange, and $r > 0.5\,\mu$m in dark blue) in November 1991, May 1992 and November 1992 at Laramie, Wyoming (41° N, 105° W) from the experiment with coagulation enhanced by the kinetic regime enhancement factor $E(\infty)$ (to be compared to the corresponding figure in the article for the experiment without van der Waals forces). Solid lines show the modelled monthly mean, while the crosses indicate the range of daily mean concentrations within that month. Optical particle counter (OPC) measurements from Deshler et al. (2003) are shown as symbols.

[Figure]

**Figure 5.** Stratospheric effective particle radius at Laramie, Wyoming (41° N, 105° W) as simulated by the LMDZ-S3A model and observed with optical particle counters (Deshler et al., 2003). The light blue (resp. orange) line shows the model result for coagulation enhanced by the continuum regime enhancement factor $E(0)$ (resp. the kinetic regime enhancement factor $E(\infty)$).

[Figure]

**Figure 6.** Evolution of the global mean stratospheric aerosol optical depth (SAOD) at 550 nm modelled with LMDZ-S3A compared to the climatology from Sato (2012) and to SAOD simulated with WACCM by Mills et al. (2016). The dashed (resp. dotted) line shows the model result for coagulation enhanced by the continuum regime enhancement factor $E(0)$ (resp. the kinetic regime enhancement factor $E(\infty)$).

3) Using prescribed chemical lifetimes is probably fine for ambient aerosol, but could be problematic with large volcanic eruptions or geoengineering. I suggest you clarify these caveats and either do some evaluations that conclude that prescribing the oxidants doesn't cause too many errors in your results, or state that this current model setup is not recommended for very large volcanic eruptions or geoengineering. Also, cite Bekki (1995) for evidence of limited oxidants. (It's reassuring that your results are reasonable for Pinatubo, but unclear whether your model is safe to use for eruptions larger than Pinatubo or for geoengineering).

We agree with the referee that assuming a constant $SO_2$ lifetime (and hence a constant OH mixing ratio) might be a limitation for very large $SO_2$ injections. Bekki (1995) showed that a constant $SO_2$ lifetime is not justified for an eruption as large as that of the Tambora, but they concluded that in the case of Pinatubo "OH changes would have been too modest to have had a noticeable effect on the global $SO_2$ removal rate". However, Bekki used a zonally-averaged 2-D model with coarse latitudinal resolution of 10°, which could not represent the large initial $SO_2$ concentration in the volcanic plume. It is thus possible that the $SO_2$ lifetime may have increased in the initial Pinatubo volcanic cloud.

In order to test the sensitivity of the results to the assumed global $SO_2$ removal rate, we performed another simulation with $SO_2$ lifetimes arbitrarily increased by a factor 2. It appears unlikely that the OH effect impacted the global $SO_2$ lifetime beyond this factor 2, notably when compared with observational studies of the volcanic $SO_2$ decay. Analyses of $SO_2$ observations after the Pinatubo eruption give a global $SO_2$ lifetime ranging from 23 to 35 days (Bluth et al., 1992; Read et al., 1993). We find that a two-fold increase in assumed $SO_2$ lifetime has a small effect on the simulated global-mean AOD following Pinatubo (Fig. 7). The peak AOD is delayed, which is more consistent with the observations, but it is also slightly decreased. We also performed a Pinatubo simulation with $SO_2$ lifetimes increased by a factor 2 on the day of the eruption and decreasing linearly to their climatological values within one month. This temporary increase in $SO_2$ lifetime delays and increases slightly the peak of the global-mean AOD. Overall the sensitivity to the $SO_2$ lifetime appears to be small.

Furthermore, there are other, probably more important, sources of uncertainties. For instance, it is likely that large amounts of water vapour were injected with $SO_2$ (Guo et al., 2004b). This water would have the opposite effect of enhancing OH levels in the $SO_2$ plume and decrease the $SO_2$ lifetime. Another uncertainty is the amount of $SO_2$ injected. There is almost a factor 2 in the possible range of $SO_2$ injection. From satellite data, Read et al. (1993) estimated an injection of 17 Tg $SO_2$ whereas another estimate reports an effective emission of 10 Tg $SO_2$ at 8-10 days after the eruption because of a possible rapid removal of $SO_2$ on ash (Guo et al., 2004a).

Finally it should be noted that many of the geoengineering stratospheric aerosol injection (SAI) scenarios envisaged consist of a continuous injection rate of $SO_2$, which is unlikely to lead to large $SO_2$ concentrations and decreased $SO_2$ lifetime. In conclusion using a prescribed chemical lifetime is probably not currently a major limitation of our model although it is desirable of course to improve the model in that respect in a future phase of model development.

The manuscript has been revised to reflect this discussion.

Specific suggestions:

Abstract: 1) Add that stratospheric aerosols impact chemistry as well as radiative forcing 2) Describe your model in more

[Figure]

**Figure 7.** Same as Fig. 6, but here the dashed (resp. dotted) line shows the model result for $SO_2$ lifetimes uniformly doubled (resp. $SO_2$ lifetimes doubled on the day of the eruption, and decreasing linearly to their climatological values within 1 month).

detail (it computes aerosol radiative heating and QBO but has semi-prescribed chemistry). 3) Remove the word "all" from relevant microphysical processes, as certainly *some* are missing, such as van der Waals forces. 4) Add a few more details of your results, e.g. number of ambient large particles are underestimated/small particles overestimated (which may be due to not including van der Waals forces).

5  We changed the first sentence to "Stratospheric aerosols play an important role in the climate system by affecting the Earth's radiative budget as well as atmospheric chemistry [...]". We also added some more details of the model and the results and removed the word "all" even though we now include the coagulation term due to van der Waals forces as an option.

Introduction Line 21: Remove the word "quite"
The word was removed.

10  2.1.1 Line 20: Does Hourdin et al. (2006) evaluate the model for stratospheric applications? If not, I suggest adding a subsection where you describe and evaluate some of the stratospheric processes in your model to ensure it is behaving reasonably, such as looking at strat age-of-air, trop-strat transport processes, seasonal/latitudinal temperatures. It is nice that you looked at QBO.

We added a reference to de la Cámara et al. (2016), where different aspects of the Brewer-Dobson circulation and the QBO in
15  the LMDZ model are evaluated and discussed.

2.1.2 Line 8: Say why you only compute nucleation, condensation etc in the stratosphere (to save computational time? or is a

different aerosol module used in the troposphere?)

LMDZ-S3A is a module developed for sulfate aerosol in the stratosphere, therefore it is only active above the tropopause. Tropospheric aerosols are treated separately in the model, either by prescribing a climatology (as is the case in the model version presented here) or by activating an interactive standard bulk aerosol model described by Escribano et al. (2016). The manuscript has been revised to explain this.

**2.1.2 Line 10: Do you compute the tropopause level at each timestep? Please clarify.**

We clarified that the tropopause pressure is computed at each timestep.

**2.2.1 Line 19: I suggest put these paragraphs in a new section called "Chemistry".**

We created a subsection called "Semi-prognostic sulfur chemistry".

**2.2.1 Line 31: Please add some discussion regarding my general concern 3) here.**

A discussion on the limitation of using prescribed chemical lifetime has been included in the revised manuscript.

**2.2.2 Line 5: The statement that coagulation and growth, not nucleation, determines particle size distribution is a main conclusion from English et al. (2011).**

We added a reference to English et al. (2011).

**2.2.5 Line 13: Please add discussion of the possible importance of van der Waals forces to accurately model ambient aerosol. Results by English et al. 2011 (see Figure 9) suggest van der Waals forces is very important to get ambient aerosol correct (likely more important for ambient aerosol than for large perturbations such as volcanic eruptions).**

As discussed above, we have introduced a parametrization of the effect of van der Waals forces on coagulation and we have performed additional sensitivity simulations. A discussion on this has been included in the revised manuscript

**2.3 Line 23: "widely tested by the authors". are there any peer-reviewed papers to cite?**

The Mie code was extensively tested. Early references describing the code and some reference calculations include Boucher (1995) and Boucher et al. (1998).

**3.1 Line 29+: (discuss the likely importance of van der Waals forces). Also, since English et al. (2011) also didn't have meteoritic dust in their model, but the size distribution looks good, the lack of meteoritic dust is not likely a source of error for particle size/mixing ratio (but probably is a source of error for burden/extinction).**

We agree that the role of meteoritic dust on the stratospheric AOD is likely to be small. However its role of the aerosol size distribution remains an open question because of a lack of reliable measurements. We have added a short discussion on this in the revised manuscript.

**3.2 Line 21: Cite Aquila et al. (2012) who also investigated the relationship between injection height and Pinatubo accuracy when aerosol heating is involved (How do your injection height results compare )**

We added a reference to Aquila et al. (2012), who also found the best agreement between the simulated and the observed sulfate cloud for an injection height of 16 to 18 km.

3.2 Line 26: Is this data published anywhere yet?

No, there is no publication on the CMIP6 stratospheric aerosol data yet. Luo Beiping provided it to the authors via personal communication.

Conclusions Line 30: Please add discussion about whether your model is safe to use for perturbations larger than Pinatubo and/or for hypothetical geoengineering schemes given that oxidants are prescribed.

As dicussed above we think the model is safe to use up to fairly large rates of SAI. The manuscript is revised to clarify this.

Figure 5 and Figure 11: It is difficult to distinguish the model and observation lines, especially the dark blue lines. Can you change the colors or markers to better distinguish?

Figure 7: Add to the caption the source of the CMIP6 aerosol data set. Please add a header to each row and column describing each panel.

We updated the respective plots and hope that the legibility improved.

**Response to comments by Referee 2**

This manuscript describes the stratospheric aerosol model S3A and it's application in the LMDZ GCM. The model allows for interaction between aerosol radiative effects and atmospheric dynamics. Comparisons with observation for a background (unperturbed) period and for the Pinatubo period are presented. The introductory section has a thorough discussion of the evolution of global aerosol models and trade-offs related to the aerosol approach chosen. The sectional approach to aerosol is appropriate for the stratosphere. The disconnect between stratospheric and tropospheric aerosols should be justified or it's limitations discussed and relegated to future model development. In particular, interactions between sulfate and organic aerosols in the UTLS region (see recent GRL paper by Yu et al. (2016)) and affects of descending sulfate aerosol on clouds and tropospheric chemistry should be mentioned. Meteoritic particles in the middle stratosphere and polar regions could also be significant in some cases. The radiative code is appropriate to the application, with 6 shortwave and 16 longwave bands. The paper is generally well-written, though a few relevant model details and caveats have been omitted, as detailed below. The model has room to grow by adding an interactive chemical scheme and strat-trop aerosol interactions, but the version presented here is still a useful contribution to the literature and worthy of documentation in Geoscientific Model Development.

We agree that the model can be further complexified but is already usable in its current form.

Specific Scientific Comments:

Page 1, lines 16-18: "Gravitational sedimentation . . . is extremely dependent on the size of the aerosol particles and ambient air density." Air density (or pressure) explains why sedimentation is not important in the troposphere but is in the stratosphere.

We agree that higher pressure partly explains why sedimentation is not important in the troposphere. But we think that the main

reason is that there are other faster processes in the troposphere that are not at work in the stratosphere. Ambient air density is a small term relative to the aerosol density. We have not modified the text here.

Page 4, lines 6-7: It is not accurate to say that "We have included processes relevant to . . . much larger and/or longer emission rates than experienced in typical volcanic eruptions" because of the prescribed oxidants converting $SO_2$ and $OCS$ to sulfate. Perturbations to OH will decrease $SO_2$ lifetime following an eruption much larger than Pinatubo. The model currently as the hooks to account for larger eruptions in the future when the REPROBUS chemical scheme is integrated into LMDZ, as explained on page 8, but currently does not have that capability.

See our reply to Referee 1 on this issue. We think that the prescribed $SO_2$ lifetime is not a significant limitation of our model unless we consider very large $SO_2$ injection rates.

Page 4, lines 23-25: Please include the height of the model top.

We added the model top height of 75 km.

Page 7, lines 24-26: Apparently the photochemical transformation of $H_2SO_4$ gas to $SO_3$ and $SO_2$ above the top of the aerosol layer is neglected? This will result in errors in nucleation in the polar regions due to downwelling of $SO_2$.

It is right that we do not photolyse $H_2SO_4$ in our model, but we do have downwelling of (previously evaporated) $H_2SO_4$ at higher latitudes, which may have a similar effect on particles there as the mentioned $SO_2$.

Page 8: lines 7-8: Wet and dry deposition are mentioned but not washout/cloud removal processes. Are aerosols removed in clouds throughout the troposphere or at the surface only?

Wet deposition in the LMDZ model includes both in-cloud and below-cloud scavenging.

Page 12, lines 17-24: Can you justify using the Steele and Hamill (1981) formulation for aerosol weight percent? How does it compare to Tabazadeh et al. (1997)?

We do not use the Steele and Hamill (1981) results. We calculate aerosol composition "following the approach described in Steele and Hamill (1981) assuming the water content of the aerosol particles to be in equilibrium with the surrounding ambient water vapour." Tabazadeh et al. (1997) used the same approach. The difference between these two studies is the expression used to calculate the partial pressure of $H_2O$ above aqueous sulphuric acid solutions. The differences in aerosol composition between these 2 studies were less than 1% at temperatures above 200 K, and hence throughout almost all the stratosphere. The only exception is in the lower stratosphere in the polar regions in winter when and where temperatures drop below 200 K. We have rephrased this part and added the Tabazadeh et al. (1997) reference: "...following the approach described in Steele and Hamill (1981) and also used in Tabazadeh et al. (1997). In this approach, the water content of the aerosol particles is assumed to be in equilibrium with the surrounding ambient water vapour."

Page 13, lines 24: How can you justify using a constant temperature and 75 weight percent aerosol in the optical properties calculation for all latitudes and altitudes? This weight percent will be pretty far off near the poles and near the top of the aerosol

 How much error does this contribute to the scattering and heating rates calculations?

We acknowledge that using constant composition in the optical properties calculation is one of the limitations of our model. It would be difficult to evaluate the error due to this assumption, but it should not be very large, as aerosol concentrations are typically low in the regions where the composition deviates considerably from the assumed value. The model may be improved

5  in this respect in the future, given that refractive index data are available for a broad range of temperatures and compositions.

Page 15-16, Pinatubo Experiment: It would be nice to see a figure showing the change in temperature, particularly near the tropical tropopause, due to the volcanic aerosols, and possibly a figure showing the change in stratospheric dynamics.

See response to Referee 1.

Minor comments:

10  Page 2, line 30: ". . .a dozen global three-dimensional stratospheric aerosol models". Omit "of"

We omitted the word.

Page 4, line 1: "Recent reviews of scientific studies. . ." Change "on" to "of"

We changed the word.

Page 6, line 20: "Nudging is activated in the model calculations described in Sect 3.2"

15  We changed the sentence accordingly.

**Changes made in the revised manuscript**

The changes in the manuscript resulting from the comments by the two Referees are marked on the following pages.

$$\lambda = \lambda_0 \cdot \left(\frac{p_0}{p}\right) \cdot \left(\frac{T}{T_0}\right) \tag{13}$$

with $\lambda_0 = 6.6 \cdot 10^{-8}\,\mathrm{m}$ for air at standard conditions $p_0 = 1013.25\,\mathrm{hPa}$ and $T_0 = 293.15\,\mathrm{K}$.

Evaporation from a particle over one time step is limited to its actual $\mathrm{H_2SO_4}$ content and condensation is limited by the available $\mathrm{H_2SO_4}$ vapour. How this is dealt with is further described in Sect. 2.2.5.

Condensation (evaporation) has an impact on the particle size distribution, shifting particles to larger (smaller) size. To account for this, we first compute the new particle volume after adding the flux $J_k(\mathrm{H_2SO_4})$ over the timestep $\Delta t$:

$$V_{k,\mathrm{new}}^{\mathrm{c/e}} = V_k \cdot \left(1 + \frac{J_k(\mathrm{H_2SO_4})\Delta t}{N_k(\mathrm{H_2SO_4})}\right) \tag{14}$$

[revised manuscript text omitted]

---

## Author Response (AR2)

**Reply to the Topical Editor Review of the revised manuscript of "The Sectional Stratospheric Sulfate Aerosol module S3A-v1 within the LMDZ general circulation model: Description and evaluation against stratospheric aerosol observations" by C. Kleinschmitt, O. Boucher, S. Bekki et al.**

Comments to the Author:

I have read the response to the reviewers comments and revised manuscript and can see that the authors have made a great deal of effort to address the reviewers comments and suggestions.
The discussion on the timescale for SO2 chemical conversion after very large volcanic eruptions, and the sensitivity experiments to test the effect of Van Der Waals forces in accelerating the rate at which particles coagulate, both represent valuable additions to the author response and the revised manuscript.
However, I found the third addition to the revised manuscript, around the discussion of the tropical stratospheric warming hard to follow and out of scope. The reviewer's motivation for suggesting the potential was to "test your aerosol radiative heating code", but the title of the paper is to assess the stratospheric aerosol simulated by the model. The warming anomaly from satellite observations represents a combination of several signals including QBO variation, simulated aerosol properties, and dynamical composition changes. That the model simulations are nudged to meteorological re-analyses fields also futher complicates matters.
I find that whole section added in the revised paper to be out of scope and request it be removed, ensuring the paper remains focussed on the topic stated in the title.
Together with this major revision, I also have two minor revisions which I strongly suggest the authors make to improve the manuscript.
Once these 3 sets of changes have been made, I expect the manuscript to be ready for publication, without requiring further referees comments. However, given the changes requested, I would ask to see the revised paper before it be allowed to proceed to publication.

1) For the reasons above, please delete the text on page 3 (lines 4 onwards) and page 4 entirely, and replace with a brief reply that given the stated topic of the paper, assessing the warming is out of scope. The additions added replying to the other two suggestions are already considerable and sufficient. In the revised manuscript, please remove the two paragraphs added to page 17.

We changed the text in the author response to the reviewers accordingly and removed the respective paragraphs from the revised manuscript.

2) The sensitivity simulations added to assess the Van Der Waals effects on coagulation are valuable additions to the paper. A minor change of emphasis is however required which I have made specific suggestion for below.
There is one minor required change to the manuscript, and then a series of minor edits to the reply to reviewers required, which are important given the wider relevance for the strong sensitivity to the coagulation rate change the results indicate, and in light of the transparent review process.

The change to the reply to reviewers, relates the statement "we have now applied the parameterization of the enhancement factor of Chan and Mozurkowich (2001)". The authors explain in the previous sentence that they have not done that, and the text needs revising here. I am specifically suggesting a series of minor wording changes to the sentences beginning "In additional sensitivity tests...." that will together achieve this. Firstly, replace "In" with "However, in" so that the sentence then follows logically from the previous one. Second, replace "have now applied the parametrization" with "test the above two limiting values of". Third, replace "(2001)" with "(2001, equations 29 and 30)" to be clear on the source of the expression. Fourth, replace "We perform two

additional simulations" with "The experiments involved". Finally, to make the rest of that sentence clearer, remove the text "a first one with coagulation enhanced uniformly by the factor E(0) and a second one" and finally replace "the factor E(infty), which is generally larger than" with "the factors E(0) and E(infty), which are 1.25 and 2.27."

We changed the text of the response accordingly, except for the last point. The values of 1.25 and 2.27 are only valid for colliding particles of identical size and for a certain temperature. But we used the size and temperature dependent parametrization of E(0) and E(infinity), which we now clarify in the text.

The text added to the actual revised manuscript requires only one minor change to rephrase "only few measurements on" with "only few measurements to constrain" (since the Van der Waals enhancement cannot be observed directly).

We changed the sentence accordingly.

3) On page 14 (1st sentence of section 3.1), state what the climatological values for COS and SO2 are that are applied as a lower boundary condition.

We clarified in the text that the climatological values for OCS and SO2 are the monthly and zonal mean values shown in Fig. 2.

On the following pages we show the revised author response to the reviewers and the revised manuscript with marked changes.

**Final author response to the Referees' comments on "The Sectional Stratospheric Sulfate Aerosol module S3A-v1 within the LMDZ general circulation model: Description and evaluation against stratospheric aerosol observations"**

Christoph Kleinschmitt[1], Olivier Boucher[2], Slimane Bekki[3], François Lott[4], and Ulrich Platt[1]

[1]Institute of Environmental Physics, Heidelberg University, Im Neuenheimer Feld 229, 69120 Heidelberg, Germany
[2]Institut Pierre-Simon Laplace, CNRS / UPMC, 4 Place Jussieu, 75252 Paris Cedex 05, France
[3]Laboratoire Atmosphères Milieux Observations Spatiales, Institut Pierre-Simon Laplace, CNRS / UVSQ, 11 Boulevard d'Alembert, 78280 Guyancourt, France
[4]Laboratoire de Météorologie Dynamique, Institut Pierre-Simon Laplace, CNRS / ENS, 24 Rue Lhomond, 75231 Paris Cedex 05, France

*Correspondence to:* Christoph Kleinschmitt (christoph.kleinschmitt@iup.uni-heidelberg.de)

We thank both Referees for their thorough evaluation of the manuscript. Their comments are repeated below in blue and our response follows in black.

**Response to comments by Referee 1**

This paper describes a new microphysical aerosol module "S3A-v1" within the LMDZ GCM for stratospheric applications, and evaluates it against ambient and volcanic observations. This is valuable research, as stratospheric aerosols are important for climate and chemistry, and can have small (ambient) impacts or large perturbations such as volcanic eruptions or hypothetical geoengineering schemes. Additionally, as the authors note, due to the long lifetime, the growth and microphysical processes are complex and important, and most modeling studies lack at least one important process. The model disclosed in this paper should be useful to advance our understanding of stratospheric aerosols. This is a well-written paper with clear structure, meaningful utility, promising results, and good grammar. I have just a couple general suggestions and numerous minor specific suggestions to be considered before publication.

We thank the Referee for their general positive appreciation.

General Minor/Moderate suggestions:

1) Additional comparisons to observations can help strengthen the paper and understand how robust your model is. A few suggestions: a) Compare vertical profiles of sulfate aerosol mixing ratio taken by aircraft (Borrmann et al. (2010); also applied in English et al. (2011) figure 10). b) Comparison to observations of sulfuric acid concentrations. (e.g. balloon data applied in English et al. (2011) Figure 5c). c) Is there UTLS and stratospheric temperature data after Mt Pinatubo eruption (to test your aerosol radiative heating code)?

The manuscript already includes a number of comparisons to observations, but we agree that comparison to additional observations can bring further insight into the model. We consider below the three suggestions made by the Referee.

a) In Fig. 1 below we compare the simulated sulfate particle mixing ratio in the tropical upper troposphere / lower stratosphere (UT/LS) to observations from aircraft reported by Borrmann et al. (2010). As already seen on Fig. 5 in the discussion paper, the comparison to optical particles counter (OPC) measurements by Deshler et al. (2003) shows that the model overestimates the number of particles significantly, mainly at the lower end of the size distribution. This positive bias is larger at higher altitudes. Said differently, the height of the maximum particle mixing ratio is shifted towards higher potential temperatures (Fig. 1). There could be several reasons for these discrepancies: the model may predict too many small particles (but sensitivity tests described below on enhancing coagulation are not entirely conclusive), the model may predict the right amount of particles but at a too high altitude, or measurements may miss a fraction of small particles. The discrepancy can also be explained by a combination of these reasons.

[Figure]

**Figure 1.** Vertical profile of sulfate particle mixing ratios measured from aircraft in the tropics under background conditions (Borrmann et al., 2010) compared to simulated profiles at the same latitudes. The measurement campaigns TROCCINOX (TR), SCOUT-O3 (SC), and SCOUT-AMMA (AM) took place in 2005 and 2006 at various tropical sites (21°S, 12°S, and 12°N, respectively). The authors would like to thank Ralf Weigel for providing them with the measurement data.

b) As it was not possible to get access to the original measurement data from Arnold et al. which are shown in Fig. 5c of English et al. (2011), we performed an eyeball comparison between English's plot and the vertical profile of $H_2SO_4$ number concentration at 43°N in September and October in LMDZ-S3A under background conditions shown in Fig. 2. The agreement between the simulated and the measured sulfuric acid concentration is very good. In both cases the highest concentrations are found at an altitude of 30–35 km and reach values of $10^7$ to $10^8$ molecules per cm$^3$, while concentrations decrease to values

of $10^5$ to $10^6$ in the lower stratosphere. Only above $\approx 40$ km our model overestimates concentrations considerably, because it does not include photodissociation of $H_2SO_4$, which causes the concentration to decrease more rapidly with increasing altitude in the observations.

[Figure]

**Figure 2.** Simulated vertical profile of $H_2SO_4$ number concentration at $43°N$ in September and October under background conditions. To be compared to measurements by Arnold et al. shown in Fig. 5c of English et al. (2011).

c) As stated in the title, this study focusses on the description of the LMDZ-S3A model and on the assessment of the simulated stratospheric aerosol. Thus, an assessment of the simulated stratospheric warming, which does not only depend on the aerosol distribution, would be out of the scope of this paper.

2) I suggest adding van der Waals forces to your coagulation scheme to improve simulations of ambient aerosol. In your Figure 5, small particles are overestimated and large particles are underestimated by about an order of magnitude in the middle/upper troposphere. Inclusion of van der Waals forces would significantly reduce this bias. This was investigated in the WACCM/CARMA model (English et al., 2011). WACCM/CARMA with van der Waals forces included in the coagulation scheme had a much better match to aerosol size distribution than your model (Fig 9 in English et al. (2011)). This was concluded to be due to van der Waals forces; the experiment without van der Waals forces overestimated particle number (Fig 10 in English et al. (2011)), similar to your model. (Neither model includes meteorites, suggesting that not including meteorites is not the problem). This should improve your comparisons in your Figure 5 and 11, and would be a nice improvement to your model. Also, you could compare vertical profiles of mixing ratio (Borrmann et al., 2010). Also, you could compare Pinatubo simulations to other studies that looked at van der Waals forces (English et al., 2013; Sekiya et al., 2016), although as you mention, the observed variability is too large to conclude with confidence whether including van der Waals forces improves volcanic simulations. Most likely, van der Waals forces is important to match ambient observations of number concentration

though.

In the original manuscript we have only considered Brownian coagulation for spherical neutral spheres (Jacobson et al., 1994). Other contributions for coagulation come from van der Waals forces, sedimentation and turbulence. Among these additional terms, only that due to van der Waals forces has been considered by some authors (English et al., 2013; Sekiya et al., 2016). Both

5    studies rely on the calculations of Chan and Mozurkewich (2001), but do not describe precisely how this additional process is treated in their models. Chan and Mozurkewich (2001) measured coagulation for sulphuric acid particles of identical size. They infer an enhancement factor over Brownian coagulation for the limit cases of the diffusion (continuum) regime ($E(0)$) and the kinetic (free molecular) regime ($E(\infty)$). These enhancement factors are not directly usable in our model because stratospheric conditions encompass both the continuum and the free molecular cases and the equations in Jacobson et al.

10   (1994) cover the general case. However, in additional sensitivity tests, we test the above two limiting cases of the enhancement factor (depending on size and temperature) of Chan and Mozurkewich (2001, equations 29 and 30) to the coagulation kernels of Jacobson et al. (1994). The experiments involved one simulation of the Pinatubo eruption with coagulation enhanced uniformly by the factor $E(0)$ and one with coagulation enhanced uniformly by the factor $E(\infty)$, which is generally larger than $E(0)$. Actual enhancement factors corresponding to stratospheric conditions can be expected to lie in between these two limit cases.

15   As anticipated by the Referee, the van der Waals coagulation term improves the comparison to observation for particle number concentration (Fig. 3) and effective radius (Fig. 4), but it makes it a little worse for AOD as shown in Fig. 5, with global-mean AOD peaking too low (and too early) compared to the Sato climatology. Given that there are only few measurements of the van der Waals coagulation term, and the mixed results obtained in our model, we do not include this process in the default version of our model, but we offer it as an option in the revised code. The manuscript has been revised to reflect this discussion.

[Figure]

**Figure 3.** Vertical profile of the cumulative aerosol number concentration (cm$^{-3}$) for three channels ($r > 0.01\,\mu$m in light blue, $r > 0.15\,\mu$m in orange, and $r > 0.5\,\mu$m in dark blue) in November 1991, May 1992 and November 1992 at Laramie, Wyoming (41° N, 105° W) from the experiment with coagulation enhanced by the kinetic regime enhancement factor $E(\infty)$ (to be compared to the corresponding figure in the article for the experiment without van der Waals forces). Solid lines show the modelled monthly mean, while the crosses indicate the range of daily mean concentrations within that month. Optical particle counter (OPC) measurements from Deshler et al. (2003) are shown as symbols.

[Figure]

**Figure 4.** Stratospheric effective particle radius at Laramie, Wyoming (41° N, 105° W) as simulated by the LMDZ-S3A model and observed with optical particle counters (Deshler et al., 2003). The light blue (resp. orange) line shows the model result for coagulation enhanced by the continuum regime enhancement factor $E(0)$ (resp. the kinetic regime enhancement factor $E(\infty)$).

[Figure]

**Figure 5.** Evolution of the global mean stratospheric aerosol optical depth (SAOD) at 550 nm modelled with LMDZ-S3A compared to the climatology from Sato (2012) and to SAOD simulated with WACCM by Mills et al. (2016). The dashed (resp. dotted) line shows the model result for coagulation enhanced by the continuum regime enhancement factor $E(0)$ (resp. the kinetic regime enhancement factor $E(\infty)$).

3) Using prescribed chemical lifetimes is probably fine for ambient aerosol, but could be problematic with large volcanic eruptions or geoengineering. I suggest you clarify these caveats and either do some evaluations that conclude that prescribing the oxidants doesn't cause too many errors in your results, or state that this current model setup is not recommended for very large volcanic eruptions or geoengineering. Also, cite Bekki (1995) for evidence of limited oxidants. (It's reassuring that your results are reasonable for Pinatubo, but unclear whether your model is safe to use for eruptions larger than Pinatubo or for geoengineering).

We agree with the referee that assuming a constant $SO_2$ lifetime (and hence a constant OH mixing ratio) might be a limitation for very large $SO_2$ injections. Bekki (1995) showed that a constant $SO_2$ lifetime is not justified for an eruption as large as that of the Tambora, but they concluded that in the case of Pinatubo "OH changes would have been too modest to have had a noticeable effect on the global $SO_2$ removal rate". However, Bekki used a zonally-averaged 2-D model with coarse latitudinal resolution of 10°, which could not represent the large initial $SO_2$ concentration in the volcanic plume. It is thus possible that the $SO_2$ lifetime may have increased in the initial Pinatubo volcanic cloud.

In order to test the sensitivity of the results to the assumed global $SO_2$ removal rate, we performed another simulation with $SO_2$ lifetimes arbitrarily increased by a factor 2. It appears unlikely that the OH effect impacted the global $SO_2$ lifetime beyond this factor 2, notably when compared with observational studies of the volcanic $SO_2$ decay. Analyses of $SO_2$ observations after the Pinatubo eruption give a global $SO_2$ lifetime ranging from 23 to 35 days (Bluth et al., 1992; Read et al., 1993). We find that a two-fold increase in assumed $SO_2$ lifetime has a small effect on the simulated global-mean AOD following Pinatubo (Fig. 6). The peak AOD is delayed, which is more consistent with the observations, but it is also slightly decreased. We also performed a Pinatubo simulation with $SO_2$ lifetimes increased by a factor 2 on the day of the eruption and decreasing linearly to their climatological values within one month. This temporary increase in $SO_2$ lifetime delays and increases slightly the peak of the global-mean AOD. Overall the sensitivity to the $SO_2$ lifetime appears to be small.

Furthermore, there are other, probably more important, sources of uncertainties. For instance, it is likely that large amounts of water vapour were injected with $SO_2$ (Guo et al., 2004b). This water would have the opposite effect of enhancing OH levels in the $SO_2$ plume and decrease the $SO_2$ lifetime. Another uncertainty is the amount of $SO_2$ injected. There is almost a factor 2 in the possible range of $SO_2$ injection. From satellite data, Read et al. (1993) estimated an injection of $17\,Tg\,SO_2$ whereas another estimate reports an effective emission of $10\,Tg\,SO_2$ at 8-10 days after the eruption because of a possible rapid removal of $SO_2$ on ash (Guo et al., 2004a).

Finally it should be noted that many of the geoengineering stratospheric aerosol injection (SAI) scenarios envisaged consist of a continuous injection rate of $SO_2$, which is unlikely to lead to large $SO_2$ concentrations and decreased $SO_2$ lifetime. In conclusion using a prescribed chemical lifetime is probably not currently a major limitation of our model although it is desirable of course to improve the model in that respect in a future phase of model development.

The manuscript has been revised to reflect this discussion.

Specific suggestions:

Abstract: 1) Add that stratospheric aerosols impact chemistry as well as radiative forcing 2) Describe your model in more

[Figure]

**Figure 6.** Same as Fig. 5, but here the dashed (resp. dotted) line shows the model result for $SO_2$ lifetimes uniformly doubled (resp. $SO_2$ lifetimes doubled on the day of the eruption, and decreasing linearly to their climatological values within 1 month).

detail (it computes aerosol radiative heating and QBO but has semi-prescribed chemistry). 3) Remove the word "all" from relevant microphysical processes, as certainly \*some\* are missing, such as van der Waals forces. 4) Add a few more details of your results, e.g. number of ambient large particles are underestimated/small particles overestimated (which may be due to not including van der Waals forces).

5  We changed the first sentence to "Stratospheric aerosols play an important role in the climate system by affecting the Earth's radiative budget as well as atmospheric chemistry [...]". We also added some more details of the model and the results and removed the word "all" even though we now include the coagulation term due to van der Waals forces as an option.

Introduction Line 21: Remove the word "quite"
The word was removed.

10  2.1.1 Line 20: Does Hourdin et al. (2006) evaluate the model for stratospheric applications? If not, I suggest adding a subsection where you describe and evaluate some of the stratospheric processes in your model to ensure it is behaving reasonably, such as looking at strat age-of-air, trop-strat transport processes, seasonal/latitudinal temperatures. It is nice that you looked at QBO.

We added a reference to de la Cámara et al. (2016), where different aspects of the Brewer-Dobson circulation and the QBO in
15  the LMDZ model are evaluated and discussed.

2.1.2 Line 8: Say why you only compute nucleation, condensation etc in the stratosphere (to save computational time? or is a

different aerosol module used in the troposphere?)

LMDZ-S3A is a module developed for sulfate aerosol in the stratosphere, therefore it is only active above the tropopause. Tropospheric aerosols are treated separately in the model, either by prescribing a climatology (as is the case in the model version presented here) or by activating an interactive standard bulk aerosol model described by Escribano et al. (2016). The manuscript has been revised to explain this.

2.1.2 Line 10: Do you compute the tropopause level at each timestep? Please clarify.

We clarified that the tropopause pressure is computed at each timestep.

2.2.1 Line 19: I suggest put these paragraphs in a new section called "Chemistry".

We created a subsection called "Semi-prognostic sulfur chemistry".

2.2.1 Line 31: Please add some discussion regarding my general concern 3) here.

A discussion on the limitation of using prescribed chemical lifetime has been included in the revised manuscript.

2.2.2 Line 5: The statement that coagulation and growth, not nucleation, determines particle size distribution is a main conclusion from English et al. (2011).

We added a reference to English et al. (2011).

2.2.5 Line 13: Please add discussion of the possible importance of van der Waals forces to accurately model ambient aerosol. Results by English et al. 2011 (see Figure 9) suggest van der Waals forces is very important to get ambient aerosol correct (likely more important for ambient aerosol than for large perturbations such as volcanic eruptions).

As discussed above, we have introduced a parametrization of the effect of van der Waals forces on coagulation and we have performed additional sensitivity simulations. A discussion on this has been included in the revised manuscript

2.3 Line 23: "widely tested by the authors". are there any peer-reviewed papers to cite?

The Mie code was extensively tested. Early references describing the code and some reference calculations include Boucher (1995) and Boucher et al. (1998).

3.1 Line 29+: (discuss the likely importance of van der Waals forces). Also, since English et al. (2011) also didn't have meteoritic dust in their model, but the size distribution looks good, the lack of meteoritic dust is not likely a source of error for particle size/mixing ratio (but probably is a source of error for burden/extinction).

We agree that the role of meteoritic dust on the stratospheric AOD is likely to be small. However its role of the aerosol size distribution remains an open question because of a lack of reliable measurements. We have added a short discussion on this in the revised manuscript.

3.2 Line 21: Cite Aquila et al. (2012) who also investigated the relationship between injection height and Pinatubo accuracy when aerosol heating is involved (How do your injection height results compare )

We added a reference to Aquila et al. (2012), who also found the best agreement between the simulated and the observed sulfate cloud for an injection height of 16 to 18 km.

3.2 Line 26: Is this data published anywhere yet?

No, there is no publication on the CMIP6 stratospheric aerosol data yet. Luo Beiping provided it to the authors via personal communication.

Conclusions Line 30: Please add discussion about whether your model is safe to use for perturbations larger than Pinatubo and/or for hypothetical geoengineering schemes given that oxidants are prescribed.

As dicussed above we think the model is safe to use up to fairly large rates of SAI. The manuscript is revised to clarify this.

Figure 5 and Figure 11: It is difficult to distinguish the model and observation lines, especially the dark blue lines. Can you change the colors or markers to better distinguish?

Figure 7: Add to the caption the source of the CMIP6 aerosol data set. Please add a header to each row and column describing each panel.

We updated the respective plots and hope that the legibility improved.

**Response to comments by Referee 2**

This manuscript describes the stratospheric aerosol model S3A and it's application in the LMDZ GCM. The model allows for interaction between aerosol radiative effects and atmospheric dynamics. Comparisons with observation for a background (unperturbed) period and for the Pinatubo period are presented. The introductory section has a thorough discussion of the evolution of global aerosol models and trade-offs related to the aerosol approach chosen. The sectional approach to aerosol is appropriate for the stratosphere. The disconnect between stratospheric and tropospheric aerosols should be justified or it's limitations discussed and relegated to future model development. In particular, interactions between sulfate and organic aerosols in the UTLS region (see recent GRL paper by Yu et al. (2016)) and affects of descending sulfate aerosol on clouds and tropospheric chemistry should be mentioned. Meteoritic particles in the middle stratosphere and polar regions could also be significant in some cases. The radiative code is appropriate to the application, with 6 shortwave and 16 longwave bands. The paper is generally well-written, though a few relevant model details and caveats have been omitted, as detailed below. The model has room to grow by adding an interactive chemical scheme and strat-trop aerosol interactions, but the version presented here is still a useful contribution to the literature and worthy of documentation in Geoscientific Model Development.

We agree that the model can be further complexified but is already usable in its current form.

Specific Scientific Comments:

Page 1, lines 16-18: "Gravitational sedimentation . . . is extremely dependent on the size of the aerosol particles and ambient air density." Air density (or pressure) explains why sedimentation is not important in the troposphere but is in the stratosphere.

We agree that higher pressure partly explains why sedimentation is not important in the troposphere. But we think that the main

reason is that there are other faster processes in the troposphere that are not at work in the stratosphere. Ambient air density is a small term relative to the aerosol density. We have not modified the text here.

Page 4, lines 6-7: It is not accurate to say that "We have included processes relevant to . . . much larger and/or longer emission rates than experienced in typical volcanic eruptions" because of the prescribed oxidants converting $SO_2$ and OCS to sulfate.

5    Perturbations to OH will decrease $SO_2$ lifetime following an eruption much larger than Pinatubo. The model currently as the hooks to account for larger eruptions in the future when the REPROBUS chemical scheme is integrated into LMDZ, as explained on page 8, but currently does not have that capability.

See our reply to Referee 1 on this issue. We think that the prescribed $SO_2$ lifetime is not a significant limitation of our model unless we consider very large $SO_2$ injection rates.

10   Page 4, lines 23-25: Please include the height of the model top.

We added the model top height of 75 km.

Page 7, lines 24-26: Apparently the photochemical transformation of $H_2SO_4$ gas to $SO_3$ and $SO_2$ above the top of the aerosol layer is neglected? This will result in errors in nucleation in the polar regions due to downwelling of $SO_2$.

It is right that we do not photolyse $H_2SO_4$ in our model, but we do have downwelling of (previously evaporated) $H_2SO_4$ at

15   higher latitudes, which may have a similar effect on particles there as the mentioned $SO_2$.

Page 8: lines 7-8: Wet and dry deposition are mentioned but not washout/cloud removal processes. Are aerosols removed in clouds throughout the troposphere or at the surface only?

Wet deposition in the LMDZ model includes both in-cloud and below-cloud scavenging.

Page 12, lines 17-24: Can you justify using the Steele and Hamill (1981) formulation for aerosol weight percent? How does it

20   compare to Tabazadeh et al. (1997)?

We do not use the Steele and Hamill (1981) results. We calculate aerosol composition "following the approach described in Steele and Hamill (1981) assuming the water content of the aerosol particles to be in equilibrium with the surrounding ambient water vapour." Tabazadeh et al. (1997) used the same approach. The difference between these two studies is the expression used to calculate the partial pressure of $H_2O$ above aqueous sulphuric acid solutions. The differences in aerosol composition

25   between these 2 studies were less than 1% at temperatures above 200 K, and hence throughout almost all the stratosphere. The only exception is in the lower stratosphere in the polar regions in winter when and where temperatures drop below 200 K. We have rephrased this part and added the Tabazadeh et al. (1997) reference: "...following the approach described in Steele and Hamill (1981) and also used in Tabazadeh et al. (1997). In this approach, the water content of the aerosol particles is assumed to be in equilibrium with the surrounding ambient water vapour."

30   Page 13, lines 24: How can you justify using a constant temperature and 75 weight percent aerosol in the optical properties calculation for all latitudes and altitudes? This weight percent will be pretty far off near the poles and near the top of the aerosol

layer. How much error does this contribute to the scattering and heating rates calculations?

We acknowledge that using constant composition in the optical properties calculation is one of the limitations of our model. It would be difficult to evaluate the error due to this assumption, but it should not be very large, as aerosol concentrations are typically low in the regions where the composition deviates considerably from the assumed value. The model may be improved in this respect in the future, given that refractive index data are available for a broad range of temperatures and compositions.

Page 15-16, Pinatubo Experiment: It would be nice to see a figure showing the change in temperature, particularly near the tropical tropopause, due to the volcanic aerosols, and possibly a figure showing the change in stratospheric dynamics.

See response to Referee 1.

Minor comments:

Page 2, line 30: ". . .a dozen global three-dimensional stratospheric aerosol models". Omit "of"

We omitted the word.

Page 4, line 1: "Recent reviews of scientific studies. . ." Change "on" to "of"

We changed the word.

Page 6, line 20: "Nudging is activated in the model calculations described in Sect 3.2"

We changed the sentence accordingly.

**Changes made in the revised manuscript**

The changes in the manuscript resulting from the comments by the two Referees are marked on the following pages.

[revised manuscript text omitted]

---

## Author Response (AR3)

**Reply to the second Topical Editor Review of the revised manuscript of "The Sectional Stratospheric Sulfate Aerosol module S3A-v1 within the LMDZ general circulation model: Description and evaluation against stratospheric aerosol observations" by C. Kleinschmitt, O. Boucher, S. Bekki et al.**

Comments to the Author:

Now that the authors have made the requested changes, I am now content for the paper to proceed to publication in GMD provided the following two very minor changes (in reply to the last set of revisions that were made in version6 of the manuscript):
1) One of my comments requested "...and finally replace "the factor E(infty), which is generally larger than" with "the factors E(0) and E(infty), which are 1.25 and 2.27."
The authors replied, "We changed the text of the response accordingly, except for the last point. The values of 1.25 and 2.27 are only valid for colliding particles of identical size and for a certain temperature. But we used the size and temperature dependent parametrization of E(0) and E(infinity), which we now clarify in the text."
The request above was based on giving the reader some indication of the size of those enhancement factors E(0) and E(infty).
Although I take the point that it would not have been correct to specify the values "are 1.25 and 2.27.", I do still feel it important to state these values -- but for the authors to add the extra few words they gave in their explanation above -- so instead to say "are 1.25 and 2.27 for colliding particles of identical size and for temperature T0" where the value of the reference temperature T0 should be given explicitly.
That will then help the reader understand the magnitude of these enhancement factors (i.e. that the Van der Walls forces can accelerate the coagulation rates by between +25% and +127%) -- for the same sized particles at that reference temperature.

We added the following sentence to the manuscript: "For colliding particles of identical size and a temperature of 298 K, E(0) and E($\infty$) have values of 1.25 and 2.27, respectively."

2) I checked the manuscript and it looks like the authors forgot to make one of the changes they agreed in their response to my Editor Review to make. See I said:
"The text added to the actual revised manuscript requires only one minor change to rephrase "only few measurements on" with "only few measurements to constrain" (since the Van der Waals enhancement cannot be observed directly).
And the authors said:
"We changed the sentence accordingly."
But I checked and that sentence still has the original wording. Perhaps it was simply a case that the authors forgot to make that change?

We apologize and now did change the sentence accordingly.

On the following pages we show the revised manuscript with marked changes.

[revised manuscript text omitted]

Flato, G., Marotzke, J., Abiodun, B., Braconnot, P., Chou, S., Collins, W., Cox, P., Driouech, F., Emori, S., Eyring, V., Forest, C., Gleckler, P., Guilyardi, E., Jakob, C., Kattsov, V., Reason, C., and Rummukainen, M.: Evaluation of Climate Models, in: Climate Change 2013: The Physical Science Basis. Contribution of Working Group I to the Fifth Assessment Report of the Intergovernmental Panel on Climate Change, edited by Stocker, T., Qin, D., Plattner, G.-K., Tignor, M., Allen, S., Boschung, J., Nauels, A., Xia, Y., Bex, V., and Midgley, P., book section 9, p. 741–866, Cambridge University Press, Cambridge, United Kingdom and New York, NY, USA, doi:10.1017/CBO9781107415324.020, 2013.

[revised manuscript text omitted]